# Persistent DNA-break potential near telomeres increases initiation of meiotic recombination on short chromosomes

Vijayalakshmi V. Subramanian [1], Xuan Zhu [2,6], Tovah E. Markowitz[1,7], Luis A. Vale-Silva [1,8], Pedro A. San-Segundo [3], Nancy M. Hollingsworth [4], Scott Keeney [2,5] & Andreas Hochwagen [1]

Faithful meiotic chromosome inheritance and fertility rely on the stimulation of meiotic crossover recombination by potentially genotoxic DNA double-strand breaks (DSBs). To avoid excessive damage, feedback mechanisms down-regulate DSBs, likely in response to initiation of crossover repair. In *Saccharomyces cerevisiae*, this regulation requires the removal of the conserved DSB-promoting protein Hop1/HORMAD during chromosome synapsis. Here, we identify privileged end-adjacent regions (EARs) spanning roughly 100 kb near all telomeres that escape DSB down-regulation. These regions retain Hop1 and continue to break in pachynema despite normal synaptonemal complex deposition. Differential retention of Hop1 requires the disassemblase Pch2/TRIP13, which preferentially removes Hop1 from telomere-distant sequences, and is modulated by the histone deacetylase Sir2 and the nucleoporin Nup2. Importantly, the uniform size of EARs among chromosomes contributes to disproportionately high DSB and repair signals on short chromosomes in pachynema, suggesting that EARs partially underlie the curiously high recombination rate of short chromosomes.

[1] Department of Biology, New York University, New York, NY 10003, USA. [2] Molecular Biology Program, Memorial Sloan Kettering Cancer Center, New York, NY 10065, USA. [3] Instituto de Biología Funcional y Genómica, Consejo Superior de Investigaciones Científicas, University of Salamanca, 37007 Salamanca, Spain. [4] Department of Biochemistry and Cell Biology, Stony Brook University, Stony Brook, NY 11794, USA. [5] Howard Hughes Medical Institute, Memorial Sloan Kettering Cancer Center, New York, NY 10065, USA. [6] Present address: Amazon AI, Seattle, WA 98101, USA. [7] Present address: Frederick National Laboratory for Cancer Research, Frederick, MD 21701, USA. [8] Present address: BioQuant Center, Heidelberg University, 69120 Heidelberg, Germany. Correspondence and requests for materials should be addressed to A.H. (email: andi@nyu.edu)

Meiosis generates haploid sex cells using two consecutive chromosome segregation events that follow a single cycle of DNA replication. To assist proper separation of homologous chromosomes in the first segregation phase (meiosis I), numerous DNA double-strand breaks (DSBs) are introduced by topoisomerase-like enzyme Spo11 to stimulate the formation of crossover recombination products (COs). Together with sister chromatid cohesion, COs connect homologous chromosome pairs and promote their correct alignment on the meiosis I spindle[1,2].

Because DSBs are potentially genotoxic, several processes choreograph DSB formation at the right place and time to maintain genome integrity[1–4]. DSBs form preferentially at hotspots that depend strongly on chromatin accessibility and appropriate histone modifications[1,5]. In addition, Spo11 activity is modulated over larger chromosomal domains by specialized chromosome architecture, in which chromatin loops emanate from a meiosis-specific protein axis (the axial element). In this architecture, DSB hotspots are primarily found on chromatin loops but are thought to translocate to the axial element to encounter proteins necessary for DSB formation[6,7]. Accordingly, mutants lacking axial-element proteins exhibit severely reduced DSB levels[8,9]. The enrichment profile of axial-element proteins correlates well with DSB levels on a broad, regional scale[6,7,10], suggesting that controlled distribution of these proteins is important for governing the distribution of meiotic DSB activity.

In addition, a network of checkpoint and feedback mechanisms controls the timing of DSB formation[2–5]. These mechanisms establish a defined window of opportunity for DSB formation by preventing DSB formation during pre-meiotic DNA replication, as well as upon exit from meiotic prophase[11–17]. Checkpoint mechanisms also suppress redundant DSB formation in the vicinity of already broken DNA[18–21]. In addition, DSBs are progressively down-regulated as prophase proceeds. Studies suggest that the synaptonemal complex (SC), an evolutionarily conserved proteinaceous structure that assembles between homologous chromosomes, is responsible for this process[16,22–24]. The SC is thought to ensure cessation of DSB formation in a chromosome-autonomous fashion and likely triggers DSB down-regulation following initiation of the obligatory CO on a given chromosome pair[16,22–26].

In *S. cerevisiae*, SC-dependent down-regulation of DSB activity is linked to the chromosomal reduction of the axis-associated HORMA-domain protein Hop1, which normally recruits DNA break machinery to the meiotic chromatin[7,23]. Reduction of chromosomal Hop1 coincides with SC assembly and depends on SC-mediated recruitment of the AAA⁺-ATPase Pch2[23,27–29]. In the absence of *PCH2*, Hop1 signal continues to accumulate on synapsed chromosomes. A similar process is observed in mouse spermatocytes[24,30]. Intriguingly, not all DSB hotspots in yeast are equally dampened upon SC assembly. A number of hotspots, including widely used model hotspots (e.g., *YCR047C* and *HIS4LEU2*–a modified hotspot at *YCL030C*), remain competent for DSB formation throughout prophase, irrespective of the presence of the SC[11,23,31]. The origin and purpose of these long-lived hotspots is unknown.

One possible function of long-lived hotspots is to increase the window of opportunity for DSB formation on short chromosomes. Short chromosomes exhibit elevated recombination density in many organisms[16,32–34]. In yeast, this bias is already apparent during DSB formation[35–38] and is likely driven by two distinct mechanisms, both of which remain poorly understood. The first mechanism enriches axis proteins and DSB factors on short chromosomes and is independent of DSB formation[7,10]. The second mechanism is thought to involve the SC-dependent down-regulation of DSBs linked to homologue engagement for DSB repair[16]. It has been proposed that shorter chromosomes are slower at engaging with their homologue, leading to prolonged DSB activity specifically on these chromosomes[16]. Accordingly, general disruption of homologue engagement leads to continued DSB formation on all chromosomes and a loss of DSB enrichment on short chromosomes[16]. This model, however, is likely incomplete because it predicts that long-lived hotspots will be restricted to short chromosomes. Instead, long-lived hotspots are also observed on long chromosomes[23].

Here, we show that most long-lived hotspots are located within large chromosome end-adjacent regions (EARs) that retain Hop1 and DSB markers in late prophase. Establishment of Hop1 enrichment in EARs requires Pch2, which preferentially removes Hop1 from interstitial chromosomal sequences, and is modulated by the histone deacetylase Sir2 and the nucleoporin Nup2. As EAR lengths are similar between chromosomes, EARs comprise a proportionally larger fraction of short chromosomes. We propose that the spatial bias in Hop1 enrichment increases relative DSB activity on shorter chromosomes and at least partially explains the increased recombination density on short chromosomes.

## Results

**Continued DSB formation is linked to chromosomal position.** To identify features distinguishing short-lived and long-lived hotspots, we expanded the number of hotspots whose lifespan has been classified using Southern blot assays. To exclude dampening of DSB activity because of prophase exit[14], we deleted the *NDT80* gene, which encodes a transcription factor necessary for initiating the prophase exit program[39]. *ndt80Δ* mutants halt meiotic progression at late prophase with fully synapsed chromosomes (pachynema) and extend the permissive time window for DSB formation[11,31], allowing more efficient capture of long-lived hotspots. Southern analysis of *ndt80Δ* cells undergoing a synchronous meiotic time course revealed additional examples of long-lived (*YOL081W*) and short-lived hotspots (*YER004W*, *YER024W*, *YOL001W*; Fig. 1a, Supplementary Fig. 1a), indicating that both hotspot classes are common in the yeast genome.

Plotting the positions of these and previously published hotspots analyzed in *ndt80Δ* mutants revealed that the differences in temporal regulation correlated closely with distance from telomeres. Whereas short-lived hotspots were located interstitially on chromosomes, long-lived hotspots were primarily found in large domains adjacent to chromosome ends (Fig. 1b). These data suggest that continued hotspot activity is linked to chromosomal position.

To extend this analysis across the genome, we assessed markers of DSB formation by ChIP-seq assay. Histone H2A phosphorylated on serine 129 (γ-H2A), the homologue of mammalian γ-H2AX, is a well-documented chromatin modification that is activated by DSB formation and spreads into an approximately 50-kb region around DNA breaks[40,41]. Samples were collected from synchronous *ndt80Δ* cultures at time points corresponding to early prophase ($T = 3$ h) and late/extended prophase ($T = 6$ h), followed by deep sequencing of γ-H2A chromatin immunoprecipitate. These analyses showed that γ-H2A is distributed relatively evenly along chromosomes in early prophase, with particular enrichment at meiotic axis sites but depletion at DSB hotspots (Supplementary Fig. 1b–d). In late prophase, however, γ-H2A enrichment was strongly biased towards the ends of all 16 chromosomes (Fig. 1c). This enrichment was most pronounced within 20–110 kb from telomeres (Fig. 1d). We refer to these regions as chromosome end-adjacent regions (EARs). Averaging across all EARs revealed that this spatial bias was also apparent in early prophase, albeit to a lesser extent (Fig. 1c (inset) and d). At both time points, γ-H2A enrichment in the EARs was above the

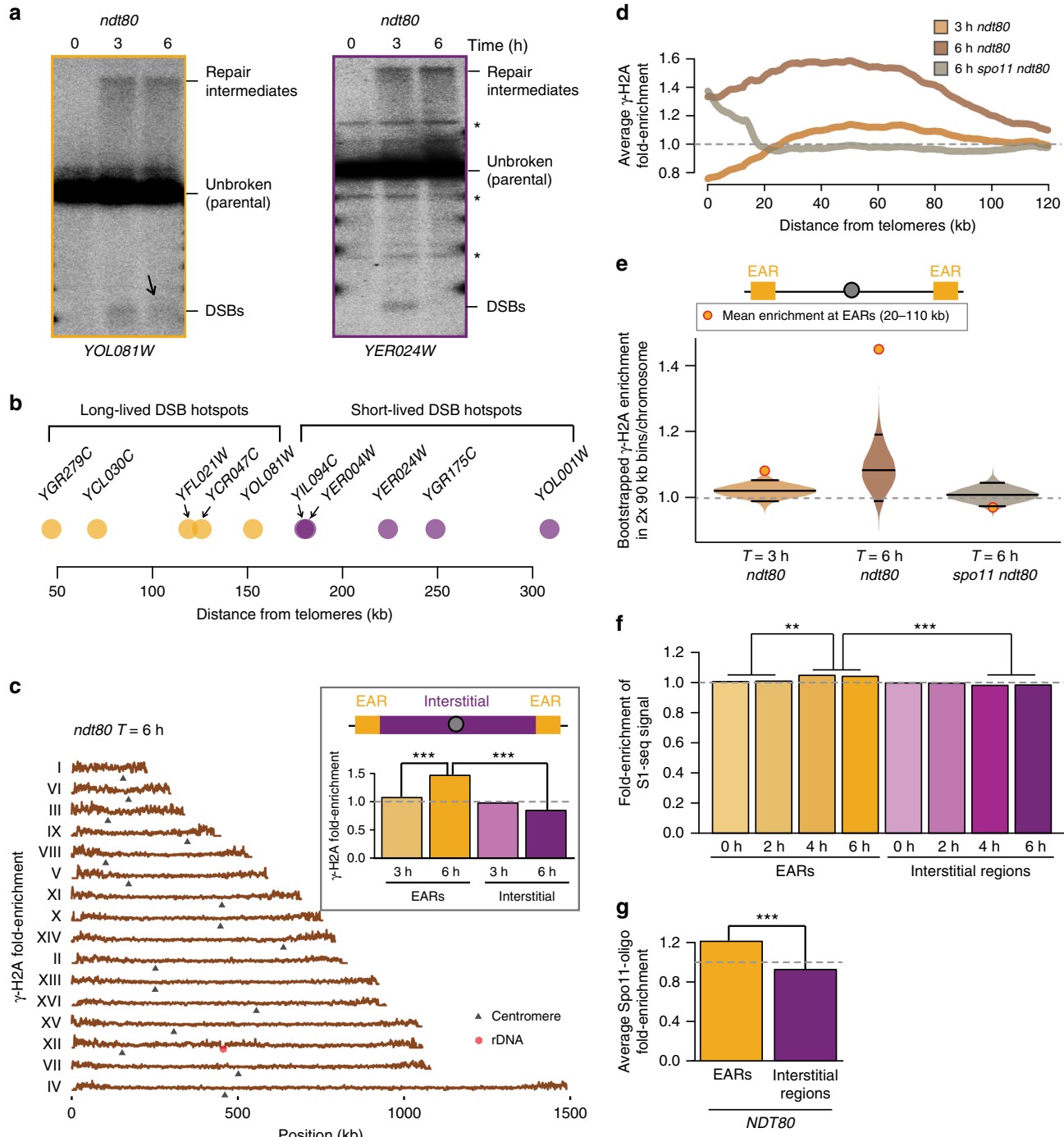

**Fig. 1** Long-lived DSB hotspots occur primarily in EARs. **a** Southern analysis of DSBs from *ndt80Δ* cells progressing synchronously though meiotic prophase at *YOL081W* (long-lived DSB hotspot, orange outline) and *YER024W* (short-lived DSB hotspot, magenta outline). Black arrow highlights continued DSBs in late prophase (*T* = 6 h) at the *YOL081W* hotspot. Asterisk (*) denotes nonspecific bands. **b** Distance of queried DSB hotspots (orange, long-lived; magenta, short-lived) from their closest telomere ([11,23,31]; this manuscript). **c** γ-H2A ChIP-seq enrichment is plotted along each of the 16 yeast chromosomes; black triangles mark the centromeres and the red hexagon marks the rDNA locus. The data are normalized to a global mean of 1. Inset shows mean enrichment at EARs (20–110 kb from telomeres; orange) and interstitial chromosomal regions (>110 kb from telomeres; magenta) in early prophase (*T* = 3 h) and extended prophase (*T* = 6 h). ***$P < 0.001$, Mann-Whitney-Wilcoxon test. **d** Mean γ-H2A enrichment in the EARs (32 domains) is plotted as a function of distance from telomeres. **e** Bootstrap-derived distributions within two 90-kb bins per chromosome from ChIP-seq data are shown as violin plots. Lower and upper quantile (95% CI), as well as the median computed from the bootstrap data are depicted as horizontal lines. The orange/red dot shows the mean ChIP-seq enrichment in EARs (20–110 kb) for the respective samples. Divergence of the boot-strap derived median from the genome average is the result of data structure and large bin sizes. **f** Time series of S1-seq signal reflecting resected DSB ends[43] are normalized to genome average and plotted as mean signal in EARs (20–110 kb, 32 domains) and interstitial chromosomal regions (>110 kb from either end of all chromosomes, 16 domains). ***$P < 0.001$ and **$P < 0.01$, ANOVA on mean enrichment followed by a post-hoc Tukey test. **g** Spo11-oligos within hotspots[16,44] are normalized to genome average and plotted as mean signal in EARs (20–110 kb, 32 domains) and interstitial chromosomal regions (>110 kb from either end of all chromosomes, 16 domains). ***$P < 0.001$, ANOVA on mean enrichment followed by a post-hoc Tukey test. The gray dashed line in **c**–**g** is the genome average

95% confidence interval (CI) of a bootstrap-derived distribution (Fig. 1e).

γ-H2A enrichment near telomeres was largely dependent on *SPO11*, indicating that these regions experience enhanced meiotic DSB activity (Fig. 1d, e). Consistently, ChIP-seq analysis of Rad51, a DSB repair protein, also showed an enrichment of signal in EARs during the extended prophase in *ndt80Δ* mutants (Supplementary Fig. 1e, f). We note that γ-H2A enrichment persisted within 20 kb from telomeres in *spo11Δ* mutants, in line with previous observations showing DSB-independent enrichment in these regions in mitotic cells[42]. These observations suggest that DSB activity in EARs is prolonged relative to genome average.

To assess DSB activity in EARs in wild-type cells (*NDT80*), we analyzed published genome-wide S1-seq datasets[43]. S1-seq measures unrepaired DNA ends and S1 nuclease-sensitive repair intermediates and thus also reports on DSB occurrence. S1-seq signal became significantly enriched in EARs over time compared to interstitial chromosomal sequences (Fig. 1f), closely mirroring the temporal enrichment of γ-H2A and Rad51 in these regions. This trend remained even after excluding the three shortest chromosomes, which consist primarily of EARs (Supplementary Fig. 1g). Analysis of available datasets[16,44] further showed that Spo11-oligonucleotides (Spo11-oligos), a byproduct of DSB formation, are also derived from EARs at significantly higher levels compared to telomere-distal regions ($T = 4$ h; Fig. 1g, Supplementary Fig. 1h). Together, these data indicate that hotspots located in EARs are partially refractory to DSB down-regulation in late prophase.

**Continued DSB formation correlates with Hop1 enrichment**. We sought to identify regulators mediating differential DSB activity in late prophase. DSB correlates well with the presence of Hop1 on meiotic chromosomes observed by immunofluorescence and genome-wide assays[7,8,10,23]. Therefore, we monitored the evolution of Hop1 enrichment on wild-type (*NDT80*) meiotic chromosomes by ChIP-seq (Supplementary Fig. 2a–c). At the time of pre-meiotic DNA replication ($T = 2$ h), Hop1 was enriched in large domains (~100 kb) around the centromeres (>95% CI; Fig. 2a, Supplementary Fig. 2a, d), likely predicting the early enrichment of Spo11 in these regions[45]. By early prophase ($T = 3$ h), Hop1 enrichment became more distributed and formed peaks of enrichment along all chromosomes (Supplementary Fig. 2b), matching previously defined sites of enrichment[10]. Importantly, by mid/late prophase ($T = 4$ h), Hop1 enrichment trended towards the EARs (Fig. 2b, Supplementary Fig. 2c). The increase of Hop1 enrichment in EARs was even more prominent in the extended prophase of *ndt80Δ*-arrested cells ($T = 6$ h; Figs 2c, d). Consistently, in both wild type and *ndt80Δ* cells, the enrichment of Hop1 in the EARs at the later time points was above 95% CI for a bootstrap-derived distribution of enrichment along the genome (Fig. 2e). To determine whether Hop1 binding increased in the EARs or decreased in the interstitial regions, we performed spike-in normalization between ChIP-seq samples (SNP-ChIP[46]). This analysis showed a 32% reduction in Hop1 binding in the extended prophase of *ndt80Δ*-arrested cells compared to the early prophase sample (Fig. 2f), indicating that Hop1 enrichment in the EARs arises primarily from a depletion of Hop1 from interstitial regions during late prophase (Supplementary Fig. 2e). These data imply that continued DSB formation results from persistent Hop1 binding in the EARs.

To test if redistribution of Hop1 in late prophase reflects an overall reorganization of the meiotic chromosome axis[27], we analyzed enrichment of the chromosome axis factor Red1 by ChIP-seq (Supplementary Fig. 3a). Similar to Hop1, Red1 enrichment in EARs became more prominent in late prophase in *ndt80Δ* mutants (Supplementary Fig. 1b). Red1 enrichment in EARs was above the 95% CI compared to a bootstrap-derived enrichment along the genome (Supplementary Fig. 3c) and significantly different from enrichment at telomere-distal regions (Supplementary Fig. 3a, inset). The persistent enrichment of axis proteins in the EARs suggests that meiotic chromatin remains poised for DSB formation in these regions during late prophase.

Because Hop1 recruits Mek1 kinase to meiotic chromosomes in response to DSB-induced checkpoint activation[47,48], we also assessed Mek1 enrichment by ChIP-seq analysis in *ndt80Δ* cells (Supplementary Fig. 3d). Mek1 was enriched along chromosomes in early prophase with specific enrichment at sites of axis-protein binding, but also at DSB hotspots (Supplementary Fig. 4a–d), consistent with Mek1-dependent histone H3 T11 phosphorylation at DSB sites[49]. Additionally, Mek1 was enriched at centromeres and tRNA genes (Supplementary Fig. 4e, f). Mek1 enrichment at axis sites persisted into late prophase, whereas enrichment at hotspots was somewhat diminished, likely reflecting a global reduction in DNA breakage in late prophase (Supplementary Fig. 4b, d). Importantly, Mek1 enrichment was significantly enhanced in the EARs in late prophase (Supplementary Fig. 3c–e), providing further support that DSBs continue to form in these regions.

**EAR-like regions flanking the ribosomal DNA**. In addition to the EARs, Hop1 enrichment in late prophase also increased in ~100-kb regions flanking the repetitive ribosomal DNA (rDNA) locus on chromosome XII (Fig. 2c, red hexagon). This increased enrichment was above the 95% CI of a bootstrap-derived distribution in *ndt80Δ* samples (Fig. 2g). A similar trend, albeit below the 95% CI, was observed in wild-type (*NDT80*) cultures. Similar to EARs, the rDNA-adjacent Hop1 enrichment was accompanied by a significant local increase in γ-H2A signal (Supplementary Fig. 5a). Mek1 enrichment also followed this trend but was below the 95% CI. To test if γ-H2A enrichment reflected continued breakage of DNA near the rDNA, we measured DSB activity at the rDNA-adjacent *YLR152C* locus using Southern analysis. Although rDNA-adjacent hotspots are generally weak[50], analysis in an *ndt80Δ* background revealed that DSBs and repair intermediates increase throughout the time course at *YLR152C* (Supplementary Fig. 5b, c), similar to long-lived hotspots in EARs (Fig. 1a). These findings demonstrate ongoing DSB activity in late prophase near the rDNA and suggest that the rDNA-adjacent regions, like EARs, escape negative feedback regulation of DSBs.

**Zip1 is equally present in EARs and interstitial regions**. Because cytological assays indicate that Hop1 is depleted from meiotic chromosomes upon SC assembly[23,27,28,51], we asked if EARs are less likely to assemble an SC than interstitial chromosomal sequences. To this end, we surveyed localization of the SC protein Zip1 on late-prophase chromosomes in *ndt80Δ* samples by ChIP-seq (Fig. 3a). Consistent with previous reports, Zip1 was enriched around centromeres (Fig. 3b). However, enrichment of Zip1 in the EARs was not different from interstitial regions ($P = 0.564$, Mann-Whitney-Wilcoxon test; Fig. 3a (inset), c, d). These findings suggest that Zip1 assembly on chromosomes is not sufficient for the spatial regulation of Hop1 in late prophase and that EARs remain enriched for Hop1 despite the presence of Zip1 in these regions.

**Pch2 is required for late-prophase EAR enrichment of Hop1**. The AAA$^+$-ATPase Pch2 is recruited to synapsed chromosomes in an SC-dependent manner and is responsible for removal of

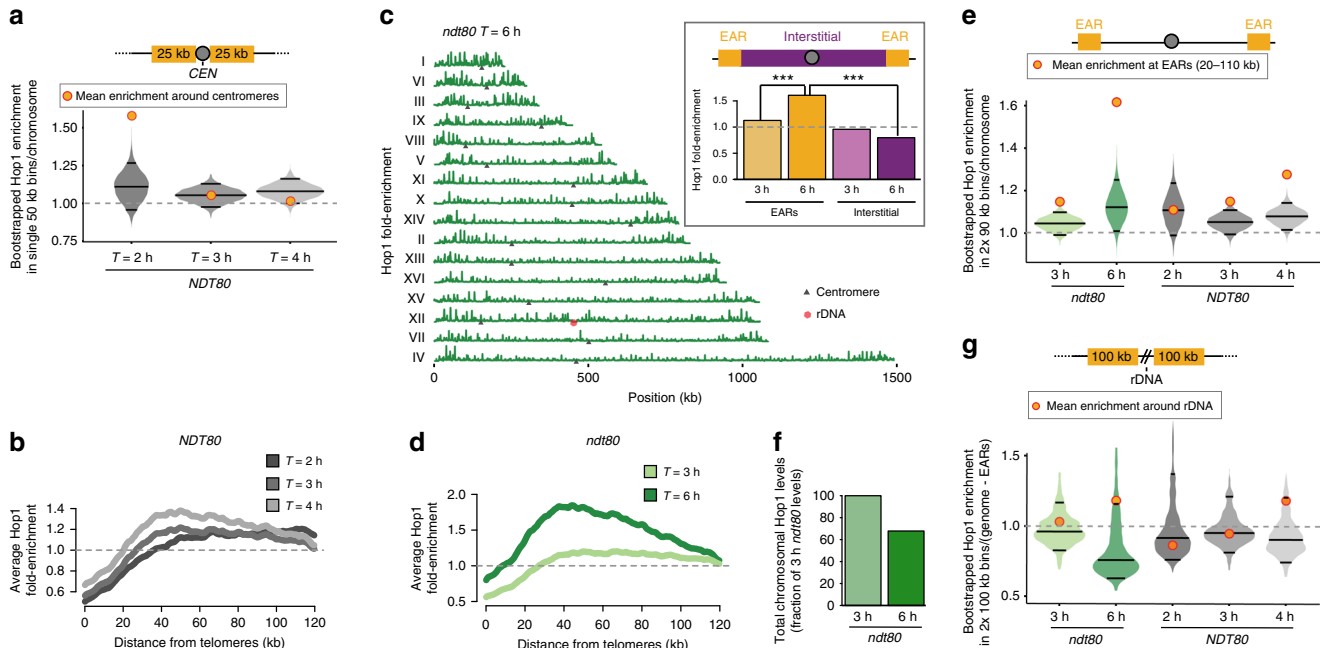

**Fig. 2** Hop1 enrichment in EARs, centromere-proximal and rDNA-proximal regions during prophase. **a** Time series of Hop1 enrichment around centromeres (50 kb) in *NDT80*. Bootstrap-derived distributions within single 50-kb bins per chromosome are shown as violin plots and the horizontal lines within the plots represent the median and the two-ended 95% CIs. The mean Hop1 ChIP-seq enrichment around centromeres (50 kb centered around centromeres) is shown as orange/red dots. **b** Mean Hop1 enrichment in the EARs (32 domains) is plotted as a function of distance from telomeres in *NDT80*. The gray dashed line is genome average. **c** Hop1 ChIP-seq enrichment (dark green) in *ndt80Δ*-arrested late prophase cells plotted along each of the 16 yeast chromosomes; black triangles mark the centromeres and the red hexagon marks the rDNA locus. The data are normalized to a global mean of 1. Inset shows mean enrichment in EARs (20–110 kb, orange) and interstitial chromosomal regions (magenta) in early prophase (*T* = 3 h) and extended prophase (*T* = 6 h). \*\*\**P* < 0.001, Mann-Whitney-Wilcoxon test. **d** Mean Hop1 enrichment in the EARs (32 domains) is plotted as a function of distance from telomeres in *ndt80Δ* cells. **e** Bootstrap-derived distributions within two 90-kb bins per chromosome from Hop1 ChIP-seq data depicted as violin plots. The horizontal lines in the violin plots represent the median and the two-ended 95% CIs. The mean Hop1 ChIP-seq enrichment in EARs (20–110 kb) for the respective samples is shown as orange/red dots. **f** Spike-in normalized total chromosomal Hop1 levels in *ndt80Δ* samples during extended prophase (*T* = 6 h) compared to early prophase (*T* = 3 h). **g** Bootstrap-derived distributions within two 100-kb bins per genome from Hop1 ChIP-seq data illustrated as violin plots. The horizontal lines in the violin plots represent the median and the two-ended CIs. The mean Hop1 ChIP-seq enrichment in rDNA-adjacent domains (100 kb on either side of the rDNA) for the respective samples is shown as orange/red dots

Hop1 from chromosomes[23,27,28]. To test if Pch2 is responsible for establishing the Hop1-enriched EARs in late prophase, we determined Hop1 binding in synchronous *pch2Δ ndt80Δ* cultures by ChIP-seq. These analyses showed pervasive binding of Hop1 along chromosomes into late prophase, consistent with the persistent cytological signal and the elevated Hop1 protein levels in *pch2Δ* mutants (Fig. 4a)[23,27,28,52]. Hop1 dropped significantly below genome average in the EARs in the absence of *PCH2* (Figs. 4b, c). Spike-in normalization between ChIP-seq samples revealed a slight reduction in Hop1 enrichment in early prophase in the *pch2Δ ndt80Δ* mutants but a 25.7% increase during the extended prophase compared to early prophase *ndt80Δ* samples (Fig. 4d). These data show that Hop1 specifically accumulates in the interstitial regions in the absence of *PCH2*, whereas binding levels in EARs remains largely unchanged. The altered Hop1 enrichment was reflected in altered DSB distribution and dynamics. Hotspots in interstitial regions continued to break and accumulate repair intermediates in *pch2Δ ndt80Δ* cultures, whereas hotspots in EARs exhibited comparatively reduced activity (Fig. 4e, Supplementary Fig. 6a, b). Consistently, Spo11-oligo counts from *pch2Δ* mutants were not significantly different between EARs and interstitial regions (*T* = 4 h; Fig. 4f). These findings suggest that Pch2 promotes Hop1 removal from interstitial regions, leading to relative Hop1 enrichment and DSB activity in the EARs in late prophase.

**Pch2 suppresses DSBs at rDNA borders and centromeres**. In addition to the broad effect on interstitial regions, several genomic landmarks were particularly affected by the loss of *PCH2*. As previously reported[50], Hop1 enrichment in *pch2Δ* mutants was increased around the rDNA array, resulting in elevated DSB levels in these regions (Supplementary Fig. 6c, d). In the extended prophase of *ndt80Δ* mutants, Hop1 ChIP enrichment was further enhanced and DSBs continued to form near the rDNA (Supplementary Fig. 6c–e).

Hop1 enrichment was also strongly elevated in the immediate vicinity of centromeres in the absence of *PCH2* (Fig. 5a). This increase was already detectable above the 95% CI in early prophase and became more pronounced in late prophase (Fig. 5b). Accordingly, Southern analysis of a centromeric hotspot (*YOL001W*) revealed elevated DSB activity in *pch2Δ ndt80Δ* mutants in late prophase (Figs. 5c, d). The increase in DSBs was not due to stalled repair because it was accompanied by an increase in repair intermediates (Fig. 5c). Likewise, Spo11-oligos were significantly elevated around centromeres (6-kb region; *P* = 0.02, Mann-Whitney-Wilcoxon test) and in the 40-kb pericentromeric regions (*P* = 0.026, Mann-Whitney-Wilcoxon test) in *pch2Δ* mutants compared to wild type (Supplementary Fig. 6f). These data indicate that Pch2 is required to restrict Hop1-linked DSB activity not only around the rDNA but also around centromeres.

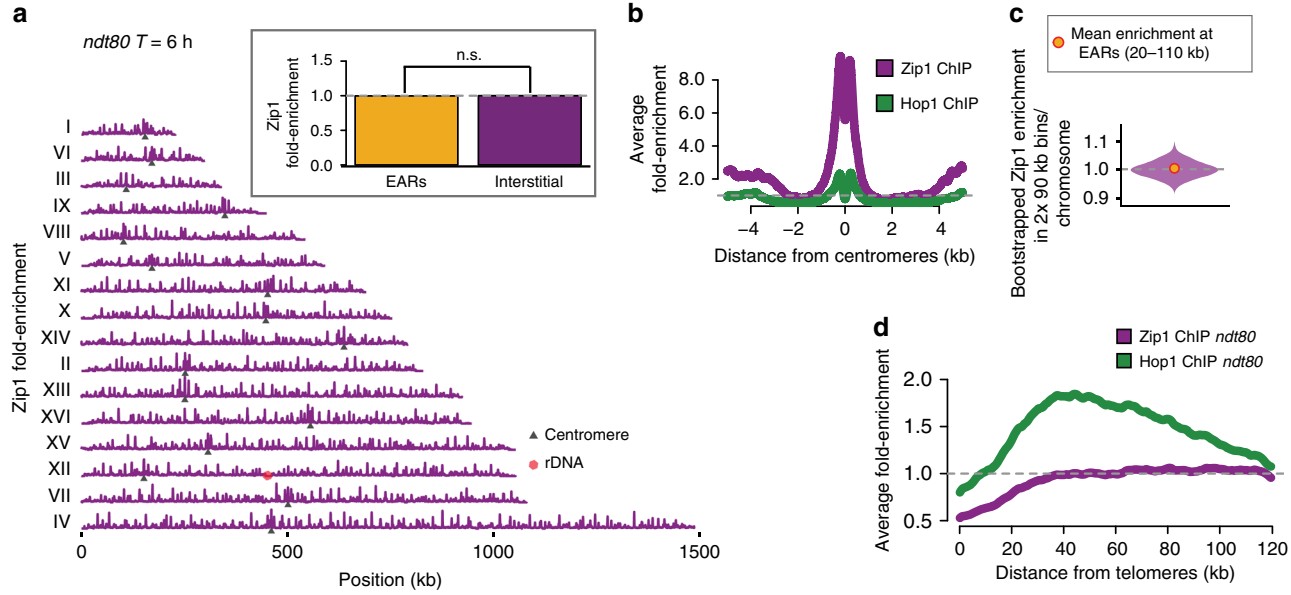

**Fig. 3** Zip1 is not depleted from EARs. **a** Zip1 ChIP-seq enrichment (magenta) in *ndt80Δ*-arrested late prophase cells plotted along each of the 16 yeast chromosomes; black triangles mark the centromeres and the red hexagon marks the rDNA locus. The data are normalized to a global mean of 1. Inset shows mean enrichment in EARs (20–110 kb, orange) and interstitial chromosomal regions (magenta) in extended prophase (*T* = 6 h). *P* = 0.564 (n.s.), Mann-Whitney-Wilcoxon test. **b** Zip1 (magenta) and Hop1 (dark green) ChIP-seq enrichment around centromeres from *ndt80Δ*-arrested cultures in late prophase (*T* = 6 h). **c** Bootstrap-derived distributions within two 90-kb bins per chromosome from Zip1 ChIP-seq data are depicted as a violin plot. The horizontal lines in the violin plot represent the median and the two-ended 95% CIs. The mean Zip1 ChIP-seq enrichment in EARs (20–110 kb) is shown as orange/red dots. **d** Hop1 (dark green) and Zip1 (magenta) ChIP-seq enrichment in late prophase (*T* = 6 h) is plotted as a mean enrichment in the EARs (32 domains) as a function of the distance from telomeres. The gray dashed line is genome average

**The nucleoporin Nup2 promotes Pch2 binding to chromosomes.** We sought to identify additional regulators that drive Hop1 enrichment in the EARs. The disruption of several telomeric factors, including the tethering factor *ESC1*, the telomere-length regulator *TEL1*, or the silencing factor *SIR3* did not significantly affect enrichment of Hop1 in the EARs (Supplementary Fig. 7a). Deletion of the meiotic telomere-clustering factor *NDJ1* severely disrupted Hop1 enrichment in EARs, while conditional depletion caused only slight effects (Supplementary Fig. 7a, b). However, the interpretation of these results is complicated by the fact that loss of *NDJ1* also causes synapsis defects[53]. As the nonessential nucleoporin Nup2 was recently identified as an interactor of Ndj1[54], we also analyzed Hop1 in *nup2Δ* mutants. Intriguingly, Hop1 did not become enriched in the EARs in late prophase in *nup2Δ ndt80Δ* mutants (Fig. 6a–c). This effect, however, was due to diminished Hop1 removal from interstitial regions, which continued to exhibit prominent Hop1 peaks in late prophase (Fig. 6a), similar to *pch2Δ* mutants. Indeed, although the phenotypes are generally less pronounced than in a *pch2Δ* mutant, loss of *NUP2* also caused retention of Hop1 on synapsed chromosomal regions (Fig. 7a, Supplementary Fig. 7c), a *pch2*-like enrichment of Hop1 in the vicinity of centromeres (Fig. 6d), and increased DSB levels at interstitial hotspots (Supplementary Fig. 6a). These data suggest that *NUP2* and *PCH2* act in a common pathway.

A common pathway is supported by the fact that the Hop1 localization defects of the single mutants are non-additive. The *nup2Δ pch2Δ* double mutant resembled the *pch2Δ* single mutant when Hop1 accumulation was analyzed on chromosome spreads (Fig. 7a, Supplementary Fig. 7c). The fact that the double mutant phenocopies the *pch2Δ* phenotype implies that *PCH2* is epistatic to *NUP2*. Consistent with this interpretation, Pch2 foci on meiotic chromosomes are noticeably diminished in the absence of

*NUP2*, with most of the Pch2 concentrated in a bright nucleolar signal (Fig. 7b). Quantification of the nucleolar signal indicated that Pch2 is even more abundant in the nucleolus in *nup2Δ* mutants than in wild type (Fig. 7c).

To investigate whether the nucleolar pool of Pch2 is functional in the absence of *NUP2*, we analyzed DSB formation and Hop1 enrichment near the rDNA. *nup2Δ* mutants do not phenocopy *pch2Δ* mutants for rDNA-associated phenotypes, as there is no DSB induction near the rDNA (Supplementary Fig. 6c, d). In fact, *nup2Δ* mutants showed a relative decrease in Hop1 ChIP-seq signal near the rDNA compared to wild type (Supplementary Fig. 6e). These results imply that the nucleolar pool of Pch2 is fully functional in the absence of *NUP2* and suggest that Nup2 is not a general activator of Pch2 function but rather acts through controlling the nuclear distribution of Pch2.

**Nup2 regulation of Pch2 is mediated by Sir2.** Nup2 may either promote the binding of Pch2 to synapsed chromosomes or suppress sequestration of Pch2 in the nucleolus. To distinguish between these possibilities, we analyzed mutants lacking the rDNA-enriched silencing factor *SIR2*, which is required for the nucleolar localization of Pch2[28]. *nup2Δ sir2Δ* double mutants showed a complete loss of Pch2 from the nucleolus (Fig. 7b, lower panels), indicating that the nucleolar over-enrichment of Pch2 in *nup2Δ* mutants is fully dependent on *SIR2*. Importantly, Pch2 signal on synapsed chromosomes was indistinguishable between *sir2Δ* and *sir2Δ nup2Δ* double mutants (Fig. 7b, Supplementary Fig. 7d), showing that *NUP2* does not promote binding of Pch2 to synapsed chromosomes. These data suggest that Nup2 counteracts Sir2-dependent recruitment of Pch2 to the nucleolus.

Consistent with this model, the aberrant Hop1 accumulation on synapsed chromosomes observed in *nup2Δ* mutants is rescued

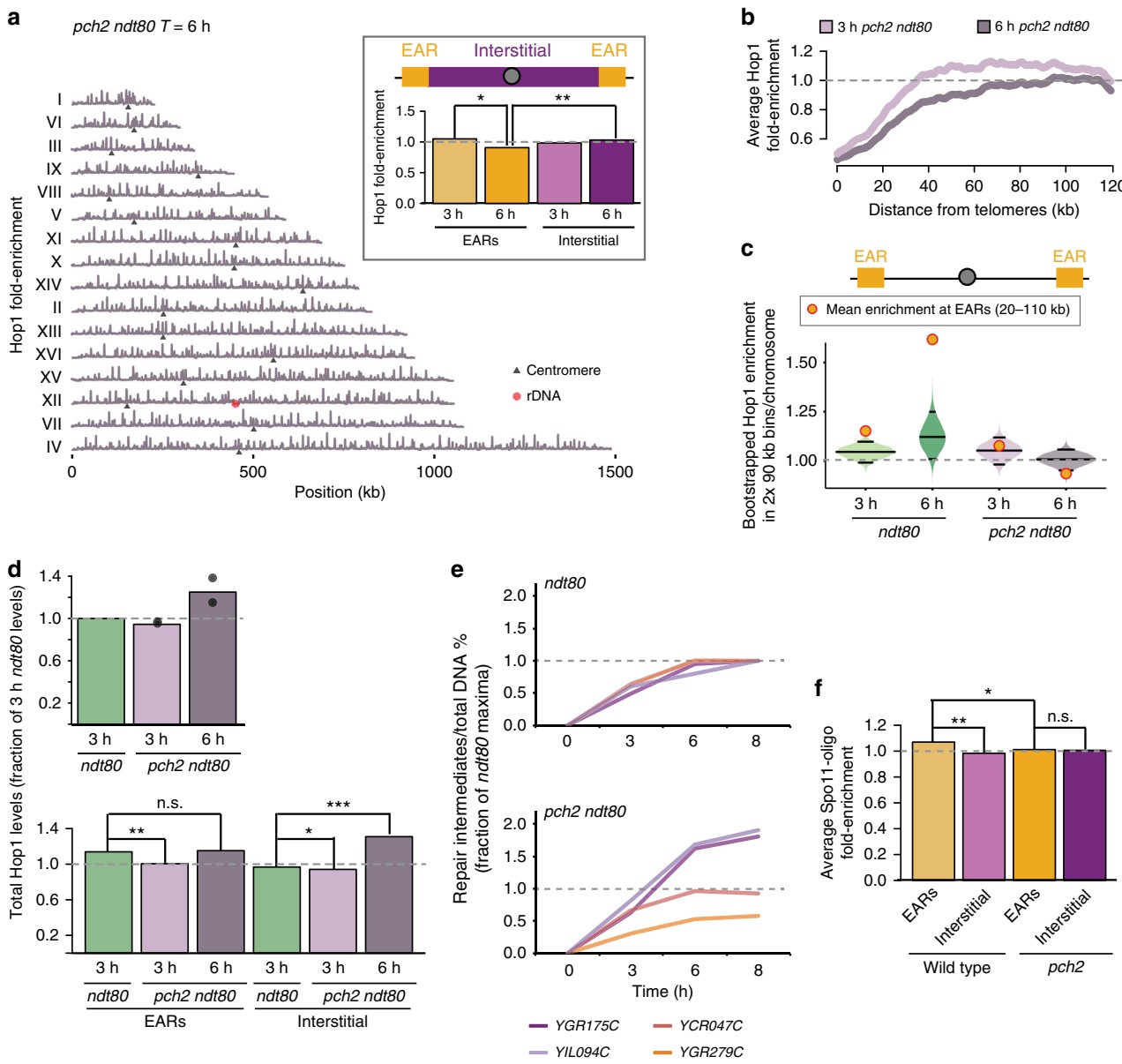

**Fig. 4** Pch2 controls regional distribution of Hop1 and DSBs in EARs. **a** Hop1 ChIP-seq enrichment in *pch2Δ ndt80Δ* late prophase (*T* = 6 h) cells plotted along each of the 16 yeast chromosomes; black triangles mark the centromeres and the red hexagon marks the rDNA locus. The data are normalized to a global mean of 1. Inset shows mean Hop1 enrichment in EARs (20–110 kb, orange) and interstitial chromosomal regions (magenta) in early prophase (*T* = 3 h) and late prophase (*T* = 6 h). **P < 0.01 and *P < 0.05, Mann-Whitney-Wilcoxon test. **b** Mean Hop1 enrichment in the EARs (32 domains) is plotted as a function of the distance from telomeres in *pch2Δ ndt80Δ* at early (*T* = 3 h) and late prophase (*T* = 6 h). The gray dashed line is genome average.
**c** Bootstrap-derived distributions within two 90-kb bins per chromosome from Hop1 ChIP-seq data illustrated as violin plots for *ndt80Δ* and *pch2Δ ndt80Δ* cells. The horizontal lines in the violin plots represent the median and the two-ended 95% CIs. The mean Hop1 ChIP-seq enrichment in EARs (20–110 kb) for the respective samples is shown as orange/red dots. The gray dashed line is genome average. **d** Spike-in normalized Hop1 enrichment in *pch2Δ ndt80Δ* at *T* = 3 h and *T* = 6 h compared to early prophase (*T* = 3 h) *ndt80Δ* sample. Total Hop1 levels are depicted in the upper panel. The range in levels is depicted with black dots. The lower panel shows spike-in normalized Hop1 levels from early prophase (*T* = 3 h) and late prophase (*T* = 6 h) *pch2Δ ndt80Δ* samples plotted as mean enrichment in EARs (20–110 kb) and interstitial chromosomal regions compared to early prophase *ndt80Δ* sample. The gray dashed line is genome average of early prophase *ndt80Δ* sample. ***P < 0.0001, **P < 0. 01, *P < 0.05, and n.s. P = 0.32, Mann-Whitney-Wilcoxon test.
**e** Cumulative DNA breaks measured as percentage of repair intermediates over total DNA and depicted as a fraction of *ndt80Δ* at *T* = 8 h are shown for *ndt80Δ*, and *pch2Δ ndt80Δ*. Interstitial hotspots (*YGR175C, YIL094C*) are shown in shades of magenta and hotspots in EARs (*YCR047C,* YGR279C) are depicted in shades of orange. The data are averages of two independent biological replicates. **f** Spo11-oligo signals from wild-type control[44] and *pch2Δ* mutant are normalized to genome average and plotted as mean signal in EARs (20–110 kb, 32 domains) and interstitial chromosomal regions (>110 kb from either end of all chromosomes, 16 domains). The gray dashed line is the genome average. **P < 0.01, *P < 0.1, n.s. not significant, ANOVA on mean enrichment followed by a post-hoc Tukey test

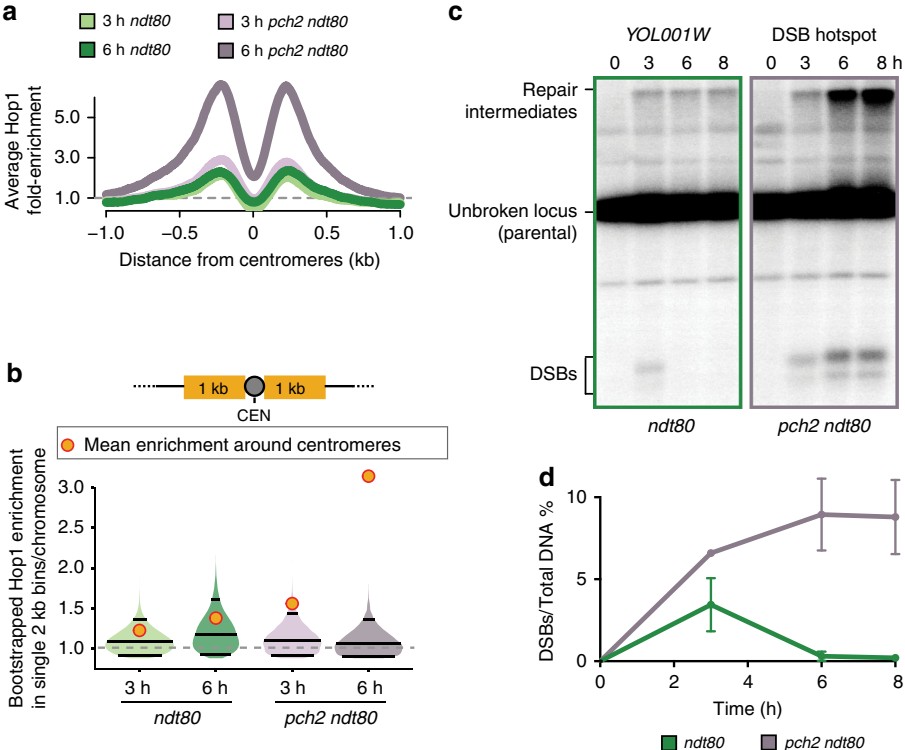

**Fig. 5** Pch2 controls regional distribution of Hop1 and DSBs around centromeres. **a** Mean Hop1 enrichment around the centromeres normalized to genome average as a function of the distance from centromeres in *ndt80Δ*, and *pch2Δ ndt80Δ*. The gray dashed line is genome average. **b** Bootstrap-derived distributions within single 2-kb bins per chromosome are illustrated as violin plots for Hop1 ChIP-seq data from *ndt80Δ*, and *pch2Δ ndt80Δ* mutants. The horizontal lines in the violin plots represent the median and the two-ended 95% CIs. The mean Hop1 ChIP-seq enrichment around centromeres (2 kb) for the respective samples is shown as orange/red dots. **c** Southern analysis of DSBs at the *YOL001W* DSB hotspot near *CEN15* in *ndt80Δ*, and *pch2Δ ndt80Δ*. Percentage of DSBs over total DNA at the *YOL001W* locus at the indicated time points is shown in **d**. The data are mean of two independent biological replicates and error bars represent the range

in the *sir2Δ nup2Δ* double mutant. Although *sir2Δ nup2Δ* mutants have substantial synapsis defects, Hop1 was never observed on synapsed chromosome fragments (Fig. 7a), suggesting efficient Hop1 removal by Pch2. These data are consistent with the abundant presence of Pch2 on chromosomes in the double mutant and indicate that *NUP2* acts upstream of *SIR2* to control Hop1 localization.

To further investigate the role of Sir2 in controlling Hop1 distribution, we performed ChIP-seq of Hop1. This analysis showed that Hop1 was precociously enriched in the EARs in *sir2Δ ndt80Δ* mutants, with strong enrichment already detectable in early prophase ($T = 3$ h) (Figs 6c, e). This pattern may reflect faster meiotic progression when Pch2 is overactive[27]. Consistent with an accelerated prophase, we noted a slightly faster appearance of fully synapsed chromosomes in *sir2Δ* mutants (Supplementary Fig. 7e), as well as faster accumulation of repair intermediates, in particular for interstitial regions (Fig. 7d). This acceleration likely reflects the premature elimination of Mek1, which normally suppresses repair[23,55]. It is possible that faster progression also leads to a premature shutdown of DSB formation because Spo11-oligo analysis at $T = 4$ h shows comparable levels of DSB formation between EARs and interstitial regions in *sir2Δ* mutant (Supplementary Fig. 7f). In line with this possibility, repair intermediates do not continue to accumulate at later time points in *sir2Δ ndt80Δ* mutants (Fig. 7d). Taken together, our data implicate a regulatory pathway consisting of Nup2, Sir2, and Pch2 in driving the enrichment of

Hop1 in EARs and controlling the window of opportunity for DSB formation in interstitial regions (Fig. 7e).

**Enrichment of Hop1 and DSB markers favors short chromosomes**. The fact that EARs occupy a proportionally larger fraction of short chromosomes (Fig. 8a) provides a possible mechanism for increasing relative DSB levels on short chromosomes. Indeed, plotting γ-H2A enrichment per kb as a function of chromosome size revealed a distinct, *SPO11*-dependent over-enrichment of γ-H2A on short chromosomes in late prophase (Fig. 8b). Similarly biased enrichment was also observed for Hop1 and Mek1 in early prophase and increased further in late prophase (Fig. 8c, Supplementary Fig. 8a). Moreover, Hop1 enrichment increased on short chromosomes during prophase regardless of whether *NDT80* was present (Supplementary Fig. 8b).

The early enrichment of Hop1 and Mek1 on short chromosomes may be driven by Red1, which recruits Hop1 to chromosomes[51,56] and also exhibits chromosome size bias for enrichment in early prophase (Supplementary Fig. 8c)[10]. Supporting this model, the pattern of chromosome size bias between Red1 and Hop1 in early prophase was not significantly different (ANOVA, $P = 0.167$). However, the Red1 chromosome size bias did not increase significantly between early and late prophase (Supplementary Fig. 8c), likely because Red1 experiences a less severe SC-dependent depletion than Hop1 in late prophase[51]. Importantly, calculating the mean Hop1 enrichment per chromosome while excluding EARs significantly reduced the

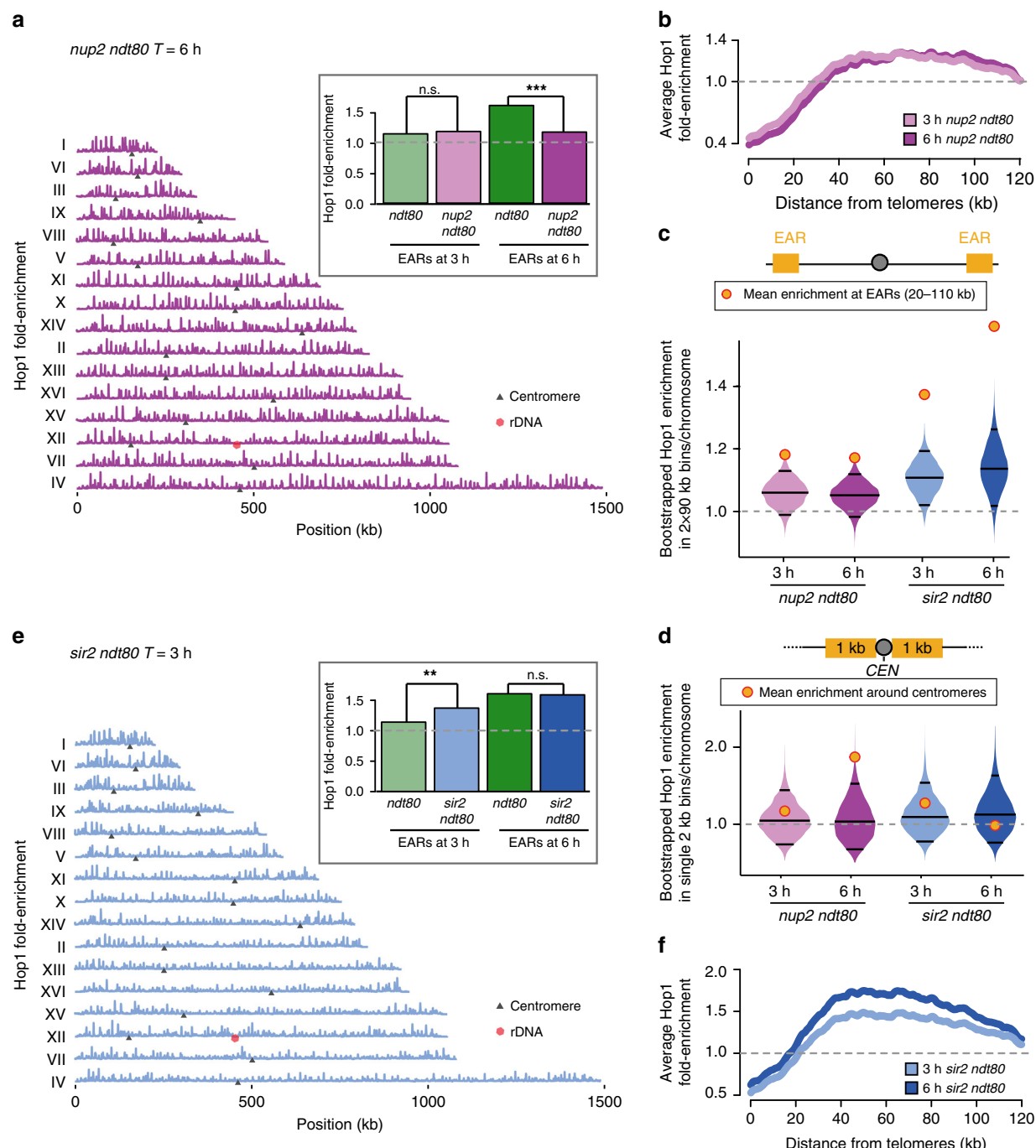

**Fig. 6** Nup2 and Sir2 also regulate regional Hop1 distribution in prophase. **a** Hop1 ChIP-seq enrichment in late prophase cells (*T* = 6 h) from *nup2Δ ndt80Δ* in magenta is plotted along each of the 16 yeast chromosomes; black triangles mark the centromeres and the red hexagon marks the rDNA locus. The data are normalized to a global mean of 1. Inset shows mean Hop1 enrichment in EARs (20–110 kb) in *nup2Δ ndt80Δ* during early prophase (*T* = 3 h, pink) and late prophase (*T* = 6 h, magenta) compared to *ndt80Δ* samples. ***\*\*\*P* < 0.001, and n.s. not significant, Mann-Whitney-Wilcoxon test. **b** Mean Hop1 enrichment in the EARs (32 domains) is plotted as a function of the distance from telomeres in *nup2Δ ndt80Δ*. **c** Bootstrap-derived distributions within two 90-kb bins per chromosome from Hop1 ChIP-seq data illustrated as violin plots for *nup2Δ ndt80Δ* (pink), and *sir2Δ ndt80Δ* (blue). The horizontal lines in the violin plots represent the median and the two-ended 95% CIs. The mean Hop1 ChIP-seq enrichment in EARs (20–110 kb) for the respective samples is shown as orange/red dots. The gray dashed line is genome average. **d** Bootstrap-derived distributions within single 2-kb bins per chromosome are illustrated as violin plots for Hop1 ChIP-seq data from *nup2Δ ndt80Δ* (pink), and *sir2Δ ndt80Δ* mutants (blue) are illustrated as violin plots. The horizontal lines in the violin plots represent the median and the two-ended 95% CIs. The mean Hop1 ChIP-seq enrichment around centromeres (2 kb) for the respective samples is shown as orange/red dots. **e** Hop1 ChIP-seq enrichment in early prophase cells (*T* = 3 h) from *sir2Δ ndt80Δ* in light blue is plotted along each of the 16 yeast chromosomes; black triangles mark the centromeres and the red hexagon marks the rDNA locus. The data are normalized to a global mean of 1. Inset shows mean Hop1 enrichment in EARs (20–110 kb) in *sir2Δ ndt80Δ* during early prophase (*T* = 3 h, light blue) and late prophase (*T* = 6 h, dark blue) compared to *ndt80Δ* samples. ***\*\*P* < 0.01, and n.s. not significant, Mann-Whitney-Wilcoxon test. **f** Mean Hop1 enrichment in the EARs (32 domains) is plotted as a function of the distance from telomeres in *sir2Δ ndt80Δ*

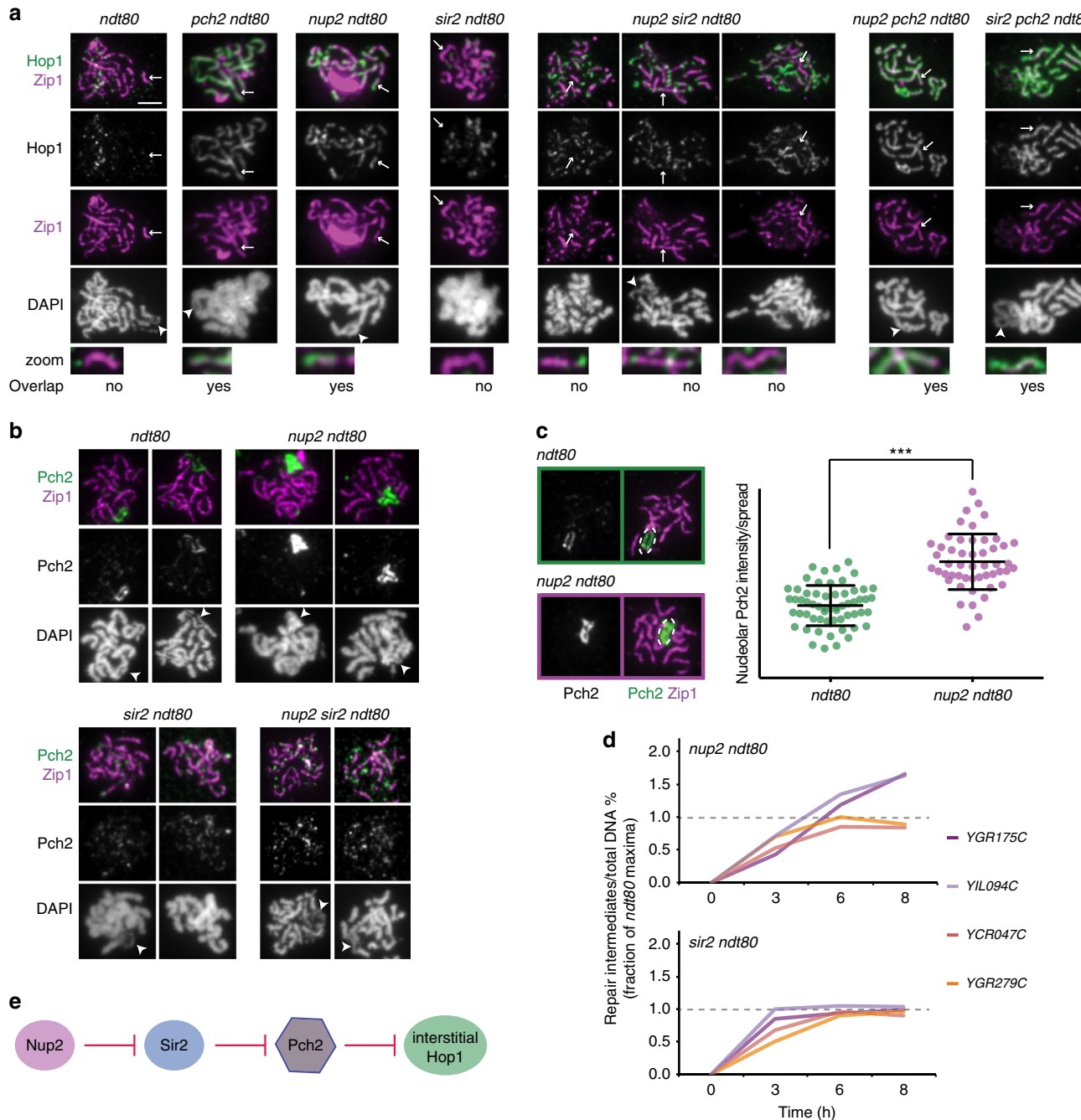

**Fig. 7** Network of Nup2, Sir2, and Pch2 regulates Hop1 and DSBs on meiotic chromosomes. **a** Representative immunofluorescence images of Hop1 (green/gray), Zip1 (magenta), and DAPI (gray) on spread chromosomes from *ndt80Δ*, *pch2Δ ndt80Δ*, *nup2Δ ndt80Δ*, *sir2Δ ndt80Δ*, *nup2Δ sir2Δ ndt80Δ*, *nup2Δ pch2Δ ndt80Δ*, and *sir2Δ pch2Δ ndt80Δ* meiotic cultures. Representative scale bar for all images shown as white line in the first panel is 2 μm. Arrow points to a synapsed region (magenta), zoomed-in signals are shown in the bottom row and presence of overlap between Hop1 and Zip1 is indicated. *n* > 40 for each genotype analyzed. In *pch2Δ ndt80Δ*, *nup2Δ ndt80Δ*, *nup2Δ pch2Δ ndt80Δ*, and *sir2Δ pch2Δ ndt80Δ* mutants, the synapsed region overlaps with Hop1 (green) but not in *sir2Δ ndt80Δ*, and *nup2Δ sir2Δ ndt80Δ* mutant samples. Arrowhead points to rDNA where easily distinguishable. **b** Immunofluorescence of Pch2 (green/gray), Zip1 (magenta), and DAPI (gray) on spread chromosomes from *ndt80Δ*, *nup2Δ ndt80Δ*, *sir2Δ ndt80Δ*, and *nup2Δ sir2Δ ndt80Δ* meiotic cultures. Arrowhead points to rDNA where distinguishable. **c** Right Panel: Quantification of nucleolar Pch2 intensity per spread nucleus in *ndt80Δ* (green dots), and *nup2Δ ndt80Δ* (purple dots). *n* > 50; error bars are S.D. from the mean. Left Panel: Representative immunofluorescence on a spread nucleus for Pch2 (green) and Zip1 (magenta) in *ndt80Δ* (top), and *nup2Δ ndt80Δ* (bottom). Measured area is outlined by dashed white ovals. **d** Cumulative DNA breaks measured as percentage of repair intermediates over total DNA and depicted as a fraction of *ndt80Δ* at *T* = 8 h are shown for *nup2Δ ndt80Δ*, and *sir2Δ ndt80Δ* mutants (see Fig. 4e). Interstitial hotspots (*YGR175C*, *YIL094C*) are shown in shades of magenta and hotspots in EARs (*YCR047C*, *YGR279C*) are depicted in shades of orange. The data are averages of two independent biological replicates. **e** Schematic representation of genetic interaction between Sir2, Nup2, and Pch2 for evicting interstitial Hop1 from meiotic chromosomes

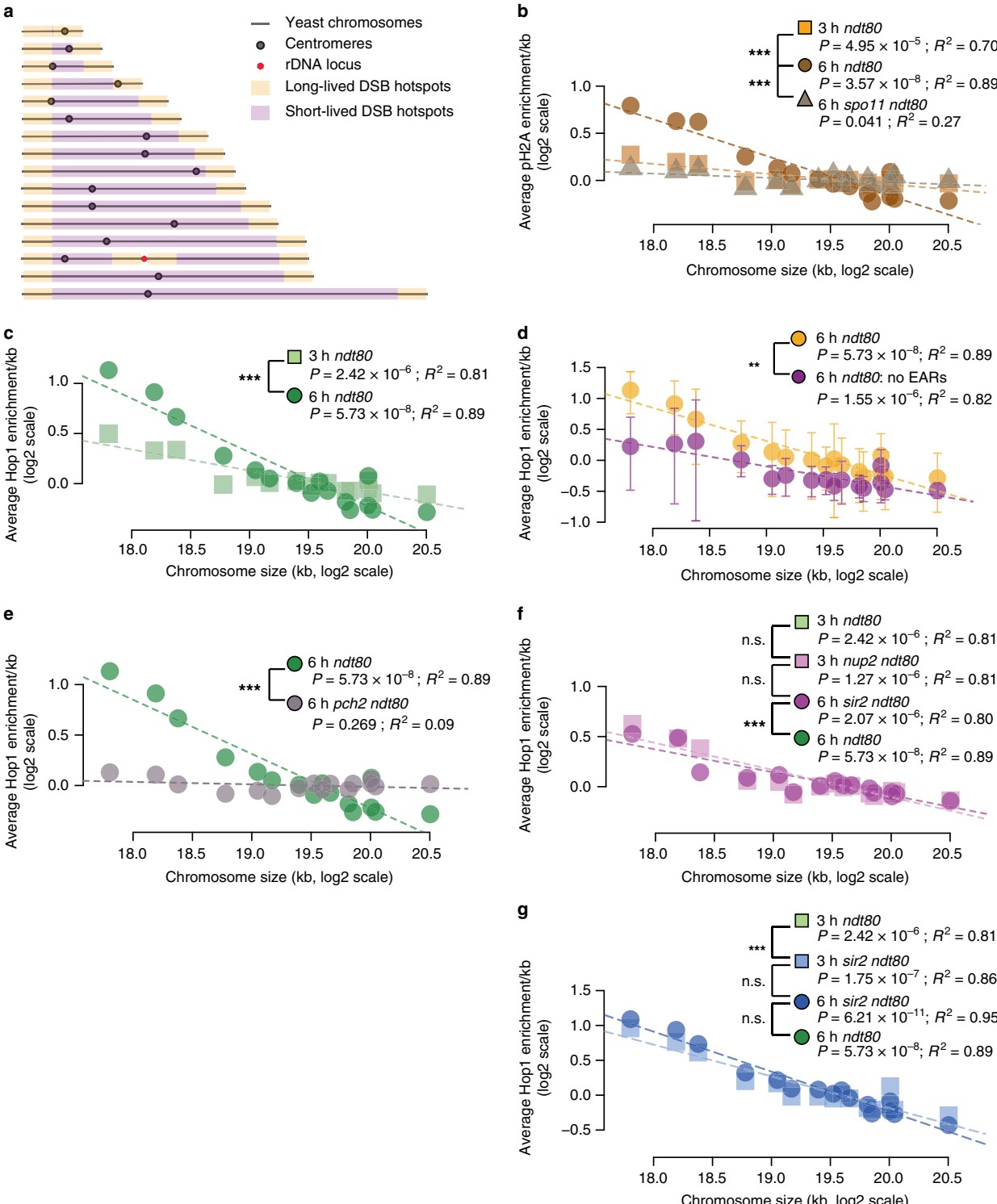

**Fig. 8** Enrichment of Hop1 and DSB markers on short chromosomes increases significantly in late prophase. **a** Schematic of telomere-adjacent enrichment of Hop1 in late prophase predicts bias for DSBs on short chromosomes. The orange bars illustrate the large domains (~100 kb) of long-lived DSB hotspots at telomere-adjacent and rDNA-adjacent regions. Interstitial regions (magenta) harbor mainly the short-lived DSB hotspots. **b-g** Mean ChIP-seq enrichment per kb is plotted for each chromosome on log scale with regression analysis. P and $R^2$ values are noted below the sample name. $R^2$, measure of the fit of the points to the line, can vary from 0 to 1.0 with 1.0 indicating a perfect fit. P is the probability of obtaining large $R^2$ values. Two-way ANOVA was performed to test significant difference in the slope between the regression lines for different ChIP-seq samples. ANOVA-derived P values are indicated, ***$P < 0.001$. γ-H2A ChIP-seq enrichment in early (T = 3 h) and late prophase (T = 6 h) in *ndt80Δ* and late prophase (T = 6 h) in *spo11Δ ndt80Δ* in **b**. Hop1 ChIP-seq in early (T = 3 h) and late prophase (T = 6 h) in *ndt80Δ* samples (**c**). Hop1 ChIP-seq in late prophase from *ndt80Δ* samples (T = 6 h) is plotted in orange (**d**). Plot in magenta is Hop1 enrichment from telomere-distal regions lacking EARs (110 kb from each chromosome end). Error bars are standard deviation of the means of 10 equal sized bins for each chromosome. Late prophase (T = 6 h) enrichment of Hop1 in *ndt80Δ* and *pch2Δ ndt80Δ* cultures (**e**). Hop1 ChIP-seq in early (3 h) and late prophase (T = 6 h) in *nup2Δ ndt80Δ* samples (**f**). Hop1 ChIP-seq in early (T = 3 h) and late prophase (T = 6 h) in *sir2Δ ndt80Δ* samples (**g**)

chromosome size bias in late prophase in both *ndt80Δ*-arrested and wild-type cells (Fig. 8d, Supplementary Fig. 8d). These analyses indicate that EARs are at least partially responsible for the late prophase enrichment of Hop1 on short chromosomes.

If EARs drive biased enrichment of Hop1 on short chromosomes then this effect should be abrogated in *pch2Δ* mutants, which do not exhibit Hop1 enrichment in EARs. Indeed, whereas the pattern of Hop1 chromosome size bias was not significantly different between *pch2Δ* and *PCH2* samples in early prophase (ANOVA, *P* = 0.691) (Supplementary Fig. 8e), in late prophase all chromosomes in *pch2Δ ndt80Δ* mutants had a similar Hop1 enrichment per kb irrespective of chromosome size (Fig. 8e). Hop1 chromosome size bias was also diminished in *nup2Δ* mutants (Fig. 8f) but enhanced in *sir2Δ* mutants (Fig. 8g). These observations indicate that a regulatory mechanism comprised of Nup2, Sir2, and Pch2 is responsible for maintaining the chromosome-size bias in meiotic prophase.

## Discussion

Our findings uncover striking regional control of DSB potential during meiosis. DSB hotspots in large domains (~100 kb) adjacent to chromosome ends, as well as regions bordering the rDNA locus, continue to break well after the SC down-regulates hotspots in interstitial chromosomal regions. This positional regulation increases the break potential on short chromosomes in the course of prophase and reveals a mechanism that promotes formation of the obligatory crossover on short chromosomes without having to measure chromosome length.

Research in several organisms, including yeast and mice, has strongly implicated the SC in mediating DSB down-regulation through the removal of HORMAD family proteins[5,16,23,24]. The results reported here are fully consistent with this view but show that, at least in yeast, chromosomes have specialized domains (EARs) that escape this down-regulation. A similar escape in regulation is also observed near the rDNA, consistent with previous observations[16]. Intriguingly, EARs retain Hop1 despite normal accumulation of the SC protein Zip1. Thus, although chromosome synapsis is spatially correlated with the removal of HORMAD family proteins[23,24,27,28], Zip1 deposition in itself is clearly not sufficient for Hop1 eviction. In *C. elegans*, which also utilizes the SC to down-regulate DSB formation in a timely manner, CO designation is thought to lead to structural changes within the SC that prevent further DSB formation[57,58]. Similar structural changes may also occur in the yeast SC and may be required for Hop1 removal[23]. If so, then Hop1 in EARs is likely protected from these effects, either because Pch2-dependent removal is suppressed or because Hop1 continues to load in these regions. The unique nature of EARs in this respect is highlighted by the fact that deletion of *PCH2* leads to a build-up of Hop1 only in the interstitial regions, while EARs become comparatively under-enriched. These data suggest that Hop1 binding in EARs is controlled by mechanisms distinct from the rest of the genome.

The molecular features that distinguish EARs from interstitial chromosomal regions remain to be discovered, although the consistent distance of EARs from chromosome ends points to a role for telomere-associated processes or the nuclear periphery. The latter possibility is particularly appealing because EAR-like domains of Hop1 enrichment are also observed near the rDNA, which is located near the nuclear envelope. We did observe effects of *NDJ1* and *NUP2* deletion on overall Hop1 enrichments patterns but our data suggest that these effects are secondary to a failure to properly synapse or recruit Pch2 activity to the interstitial regions, respectively. Analysis of a limited set of additional telomeric regulators (*TEL1*, *SIR3*) and a nuclear envelope factor (*ESC1*) did not yield regulators of EAR establishment. These

results obviously do not exclude redundant mechanisms or a role for different telomeric or nuclear-envelope factors. However, it is equally possible that other dynamic chromatin features, such as differences in replication timing, gene activity, or chromatin topology govern the Hop1 enrichment and provide the architectural basis of EARs.

We also observed unexpected dynamics of Hop1 around centromeres. Most notably, we found a strong centromeric enrichment of Hop1 in the earliest stages of prophase, before Hop1 has fully accumulated on chromosome arms. Centromeric Hop1 enrichment may similarly reflect nuclear architecture because prior to the tethering of telomeres to the nuclear envelope, the centromeres are clustered at the spindle pole body embedded in the nuclear envelope[59]. Interestingly, Spo11 is also enriched near centromeres before distributing to the arms[45] and is likely active in these regions because we observe centromeric DSBs in early prophase (see Fig. 5c, left panel). Curiously, Mek1 is also enriched around centromeres. Centromeric recruitment of Mek1 is unexpected because Mek1 suppresses repair with the sister chromatid[23,48,60], yet DSBs at centromeres are thought to be channeled by Zip1 to primarily use the sister for recombination to protect against chromosome missegregation[61,62]. Perhaps, Zip1 suppresses Mek1 activity at the centromeres without evicting it. Another unexpected finding is that Pch2 suppresses DSBs around centromeres in late prophase. However, as COs are not enhanced around the centromeres in *pch2Δ* mutants[63], Zip1 activity must be sufficient to prevent any deleterious inter-homologue COs in this situation. These findings indicate that several mechanistic layers restrict DSBs and COs at centromeres, highlighting the importance of limiting COs in these regions.

Our findings also offer insights into the regulation of Pch2. We show that the nucleoporin Nup2 promotes the binding of Pch2 to synapsed chromosomes, and we provide genetic evidence that Nup2 functions by counteracting the histone deacetylase Sir2. This regulation may not be direct as Nup2 is a component of the nuclear pores, whereas Sir2 and Pch2 are strongly enriched in the nucleolus. We note, however, that Sir2 is also present at euchromatic replication origins[64], which may also be sites of Pch2 activity[50]. Furthermore, Nup2 is a mobile nucleoporin that interacts with chromatin to regulate transcription and contributes to boundary activity[65,66], which may allow for interactions with Sir2 and the regulation of Hop1 on chromosomes.

Our analysis sheds important light on the mechanistic basis of the meiotic chromosome-size bias for recombination. In several organisms, including humans, short chromosomes exhibit higher levels of recombination[32–34,63,67], a bias that in yeast is already apparent from elevated levels of axis protein deposition and DSB formation[16,35–38]. As EARs are of similar length regardless of chromosome size, EARs comprise a proportionally much larger fraction of short chromosomes (Fig. 8a). The bias in Hop1 enrichment could thus mediate the establishment of chromosome size bias in DSBs and COs.

Available data suggests that chromosome size bias for Hop1 enrichment in late prophase is a direct consequence of preferential SC-dependent removal of Hop1 from the interstitial regions. Consistent with this notion, disrupting either CO-associated SC assembly (by deleting the CO-implementing factor *ZIP3*) or preventing the SC from removing Hop1 (by deleting *PCH2*) leads to a loss of chromosome size bias for recombination[16,63,68]. Intriguingly, in both situations, the failure to remove Hop1 differentially affects the EARs. In *zip3Δ* mutants, DSB enrichment in EARs is diminished compared to wild type (*P* = 0.58, Mann-Whitney-Wilcoxon test; Supplementary Fig. 8f). Similarly, the percentage of COs and noncrossovers (NCOs) per meiosis drops significantly in the EARs of *pch2Δ* mutants, while average CO (and NCO) counts per chromosome surge with

increasing chromosome size[63]. The persistence of Hop1 and Mek1 in the EARs likely also differentially affects repair progression[23]. This regulation could explain the apparent increase in resection tract lengths observed on short chromosomes in late prophase[43]. Although short chromosomes in yeast are slowest to synapse[69], we find no evidence of reduced Zip1 accumulation on short chromosomes during the extended prophase of ndt80Δ cells (Supplementary Fig. 8g). These data suggest that synaptic delays are too small to be detected by ChIP-seq analysis and therefore cannot explain the elevated Hop1 levels on short chromosomes observed by the same assay.

COs are enriched in sub-telomeric regions in several organisms[32,70–74]. Recent work indicated that sub-telomeric regions in barley are the first to initiate recombination, leading to a relative depletion of recombination events in the interstitial regions[75]. This temporal pattern in recombination competence is opposite to the pattern reported here for S. cerevisiae, but the ultimate outcome of increased subtelomeric recombination is similar. Available data in yeast and humans is consistent with a telomere-guided effect in establishing these regions. Chromosome bisection caused a local increase in DSB levels in yeast[35]. Moreover, an ancient telomeric fusion that gave rise to human chromosome 2 led to a decrease in crossovers rates near the fused chromosome ends compared to chimpanzees, which maintained the two separate chromosomes[70]. Thus, some features of EARs may well be evolutionarily conserved.

We propose that EARs provide a safety mechanism that ensures that DSB formation is not prematurely inactivated by the formation of the SC. Premature down-regulation of the DSB machinery is particularly problematic for short chromosomes because of their inherently lower number of DSB hotspots. By establishing privileged regions that are refractory to this down-regulation, cells may ensure that all chromosomes retain a (limited) potential for DSB formation and successful crossover recombination throughout meiotic prophase.

## Methods
**Method details.** Growth conditions: Synchronous meiotic time-courses were performed in the following manner[23]. The strains were first patched on glycerol media (YPG) and then transferred to rich media with 4% dextrose (YPD 4%). The cells were then grown at 23 °C for 24 h in liquid YPD and diluted into pre-sporulation media (BYTA, 1% yeast extract, 2% bactopeptone, 1% potassium acetate, 50 mM potassium phthalate) at $A_{600}$ 0.3. The BYTA culture was grown at 30 °C for 16 h. The cells were washed twice in sterile water and transferred to sporulation media (0.3% potassium acetate) at 30 °C to induce synchronous sporulation. Samples for ChIP-seq (25 mL) or DSB Southern assays (10 mL) were collected at the indicated timepoints. Growth conditions for obtaining Spo11-oligo sequences were as follows[44]. Cells were grown at 30 °C in 1% yeast extract, 2% bactopeptone, 1% potassium acetate for 14 h and sporulated in 2% potassium acetate.

**Experimental model and subject details.** All strains used in this study are listed in Supplementary Table 1 and are in the SK1 background except where noted.

**DSB Southern analysis.** Meiotic cells collected at the indicated time points were embedded in agarose plugs to minimize background from random shearing and genomic DNA was extracted[50]. The plugs were washed 4× in TE (Tris/EDTA) for 1 h each wash followed by 4× 1 h washes in the appropriate NEB buffer (New England Biolabs). Plugs for each time-point were transferred to separate tubes and melted at 65 °C. The genomic DNA in molten agarose was equilibrated at 42 °C prior to incubation with appropriate restriction enzyme(s). The digested DNA was electrophoresed in 0.8% agarose (Seakem LE) in 1× TBE (Tris/Borate/EDTA) at 2.6 V cm⁻¹ for 18 h. The DNA was transferred to a Hybond-XL nylon membrane (GE Healthcare) by capillary transfer and detected by Southern hybridization[23]. Restriction enzymes used to fragment genomic DNA for DSB analysis, as well as the primer sequences to construct probes are listed in the Supplementary Table 2. Probes labeled with ³²P dCTP were generated using the listed primers and a Prime-It random labeling kit (Agilent, catalog# 300392). Southern blots were exposed on an Fuji imaging screen and the phospho-signal was detected on Typhoon FLA 9000 (GE) and quantified using ImageJ software 'http://imagej.nih.gov/ij/'. In ndt80Δ samples, repair intermediates provide a

cumulative view of all DSB occurrence. Thus, in Southerns where DSB signals were faint (Supplementary Fig. 5b), the sum of DSBs and repair intermediates was quantified. Only repair intermediates were quantified in Figs 4e, 7d, because DSB signals were very faint and could not be quantified reliably. Plots were generated using the Graphpad program in Prism.

**Mechanical spreading of chromosomes and immunofluorescence.** Meiotic cells collected at indicated timepoints were treated with 200 mM Tris pH7.5/20 mM dithiothreitol (DTT) for 2 min at room temperature. Cells were spheroplasted at 30 °C in 2% potassium acetate/1 M sorbitol/0.13 µg/µL zymolyase T100. The spheroplasts were gently washed and resuspended in ice-cold 0.1 M MES pH6.4/1 mM EDTA/0.5 mM MgCl₂/1 M sorbitol. Two volumes of 3% para-formaldehyde/3.4% sucrose was added to spheroplasts on clean glass slide (soaked in ethanol and air-dried) immediately followed by four volumes of 1% Lipsol. The contents were mixed by gently tilting the slide. After at least 1 min, four volumes of 3% para-formaldehyde/3.4% sucrose was added to the samples. The chromosome spreads were then made by mechanically spreading the samples with a glass rod. The samples were rinsed in 0.4% Photoflo (Kodak, catalog# 1464510), dried overnight at room temperature and stored at −80 °C.

Immunofluorescence on the slides was performed with the desired primary antibodies followed by fluorescent-conjugated secondary antibodies listed in Supplementary Table 3. Deltavision Elite imaging system (GE) equipped with Olympus ×100 lens/1.40 NA UPLSAPO PSF oil immersion lens and an InsightSSI Solid State Illumination module was used to image samples. Images were captured using the Evolve 512 EMCCD camera in the conventional mode and analyzed with softWoRx 7.0.0 software. SoftWoRx 7.0.0 software was also used to quantify signal. DAPI was used to identify the chromosomal signal. An oval was created around the nucleolar Pch2 signal (which is more intense than nuclear Pch2) to quantify nucleolar Pch2 in Fig. 7c. Scatterplots were generated using Graphpad program in Prism and statistical significance was determined using Mann-Whitney-Wilcoxon test.

**Chromatin immunoprecipitation and Illumina sequencing.** Twenty-five milliliter of sample was collected from sporulation cultures at the indicated time points and crosslinked in 1% formaldehyde (Sigma) for 30 min. Formaldehyde was quenched with 125 mM glycine. ChIP was performed by lysing the cells in 500 µl of lysis buffer (50 mM HEPES/KOH pH 7.5, 140 mM NaCl, 1 mM EDTA, 1% Triton X-100, 0.1% sodium deoxycholate) with protease inhibitors (1 mM PMSF, 1 mM Benzamidine, 1 mg ml⁻¹ Bacitracin, one Roche Tablet (catalog# 11836170001) in 10 ml) and glass beads in a biopulveriser[76]. The sample was sonnicated to obtain DNA at an average length of 500 bp. Immunoprecipitation was performed on clarified sample using antibodies listed in the Supplementary Table 3. Libraries for ChIP sequencing were prepared by PCR amplification using Illumina TruSeq DNA sample preparation kits v1 and v2 but adaptors were used at 1:20 dilution[10]. Quality of the libraries was checked on 2100 Bioanalyzer or 2200 Tapestation. Libraries were quantified using qPCR prior to pooling. The ChIP libraries were sequenced on Illumina HiSeq 2500 or NextSeq 500 instruments at the NYU Biology Genomics core to yield 51/50 bp single-end reads. For spike-in normalization (SNP-ChIP), SK288c crosslinked meiotic cells were added to respective samples at 10–20% prior to ChIP processing[46]. SNP-ChIP libraries were sequenced on NextSeq 500 to yield 100 bp single-end reads. All ChIP-seq data sets are listed in Supplementary Table 4.

**Processing of reads from Illumina sequencing.** Illumina output reads were processed in the following manner[77]. The reads were mapped to SK1 genome[78] using Bowtie[79]. SNP-ChIP library reads were aligned to concatenated genome assemblies of SK1 and S288c genomes[46]. Only reads that mapped to a single position and also matched perfectly to the reference genome were retrieved for further analysis. 3' ends of the reads were extended to a final length of 200 bp using MACS2 2.1.1 'https://github.com/taoliu/MACS' and probabilistically determined PCR duplicates were removed. The input and ChIP pileups were SPMR-normalized (signal per million reads) and fold-enrichment of ChIP over input data was used for further analyses. The pipeline used to process Illumina reads can be found at 'https://github.com/hochwagenlab/ChIPseq_functions/tree/master/ChIPseq_Pipeline_v3/'. The pipeline used to process SNP-ChIP reads and calculate spike-in normalization factor can be found at 'https://github.com/hochwagenlab/ChIPseq_functions/tree/master/ChIPseq_Pipeline_hybrid_genome/ '.

**Spo11-oligo mapping.** Spo11-oligos were immunoprecipitated from 50 ml of synchronous meiotic culture ($T = 4$ h samples) with 80 µl of the anti-FLAG antibody in the first round and 25 µl in the second round of immunoprecipitation[44]. The Spo11-oligos were sequenced on Illumina HiSeq in the Memorial Sloan Kettering Cancer Center (MSKCC) Integrated Genomics Operation core facility. The adaptors were clipped followed by alignment of the oligo reads to S288c (sacCer2) reference genome using a custom pipeline[16,38]. Averaged maps from biological replicates were used for further analysis. Oligos within the rDNA (coordinates 451,000 and 471,000 on ChrXII) are highly enriched in sir2Δ datasets and were removed from all datasets prior to analysis. Spo11-oligo datasets analyzed in this manuscript are listed in Supplementary Table 5.

**Quantification and statistical analyses**. ChIP-seq data from two biological replicates were merged prior to analyses using the ChIPseq_Pipeline_v3 except for Hop1 ChIP from *sir3Δ ndt80Δ*, *tel1Δ ndt80Δ*, *ndj1Δ ndt80Δ* samples in Supplementary Fig. 7a and No tag *ndt80Δ*, and *NDJ1-FRB ndt80Δ* samples for Ndj1 depletion studies in Supplementary Fig. 7b for which replicate experiments were not performed. All datasets were normalized to global mean of one and regional enrichment was calculated. The R functions used can be found at 'https://github.com/hochwagenlab/hwglabr2/' and 'https://github.com/VijiSubramanian/chrEnds'.

Statistical significance tests were performed in R 3.3.3. Either Mann-Whitney-Wilcoxon test on mean enrichment by chromosome (16 chromosomes; EARs versus interstitial) was used to test for significance or one-way ANOVA with post-hoc Tukey test was used. Two-way ANOVA for multiple linear regressions with interaction was performed on log2-scaled ChIP-seq enrichment to test variation in the slopes (chromosome size bias) of two different samples. For bootstrap analyses, random samplings of the ChIP data were performed on each of the 16 circularized chromosomes and this was repeated 5000 times. The samplings were equivalent to the experimental query in size and number for each experiment. Specifically, bootstrap samplings for EARs were two unlinked samplings from each of the 16 chromosomes, for the centromeres bootstrap involved only a single sampling of each of the 16 chromosomes while for the rDNA bootstrap the entire genome was sampled twice. Additionally, to assay enrichment at the rDNA borders, EARs (120 kb from either telomere) were excluded from the genome for random bootstrap samplings. Both averaged random sampling data and experimental query were normalized to genome average. The median and two-sided 95% CI was calculated based on the spread of the bootstrap-derived distribution of enrichment.

**Reporting summary**. Further information on experimental design is available in the Nature Research Reporting Summary linked to this article.

## Data availability

The source data underlying all Figures and Supplementary Figures are provided as a Source Data file. All datasets reported in this paper (except published datasets) are available at the Gene Expression Omnibus (GEO) with the accession number 'GSE105111' for ChIP-seq datasets and 'GSE122882' for Spo11-oligo datasets. All data is available from the authors upon reasonable request. Genome-wide S1-seq datasets (Supplementary Table 6) for wild-type meiosis, GEO accession number GSE85253, were obtained from ref. [43]. The processed dataset aligned to the S288c reference genome (sacCer2) was used. The mapped Spo11-oligo counts within hotspots for wild type and *zip3Δ* mutant, GEO number GSE48299, aligned to the S288c reference genome (sacCer2) were obtained from ref. [16]. Wild type datasets from GEO dataset GSE67910[44] were used as control for Spo11-Flag oligo sequencing data from *pch2Δ* and *sir2Δ* samples.

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

## Acknowledgements

We are grateful to Brian Parker at NYU for advice on statistics. We thank the NYU Genomics Core facility for technical assistance and data processing. This work was funded in part by grant R01 GM111715 from the NIH and research grant #6-FY16–208 from the March of Dimes Foundation to A.H.; grant R35 GM118092 from the NIH to S.K. and MSKCC Cancer Center Core Grant P30 CA008748; grant R01 GM050717 from the NIH to N.M.H.; grants BFU2015–65417-R from MINECO and CSI084U16 from Junta de Castilla y León in Spain to P.A.S.

## Author contributions

Conceptualization, V.V.S. and A.H.; Investigation, V.V.S., X.Z., S.K. and A.H.; Software, V.V.S., T.E.M., L.A.V.S. and A.H.; Formal analysis, V.V.S. and A.H.; Resources, V.V.S., P.A.S, N.M.H. and A.H.; Writing–original draft, V.V.S. and A.H.; Writing–review and editing, V.V.S., X.Z., T.E.M., L.A.V.S., P.A.S, N.M.H., S.K. and A.H.

## Additional information

**Competing interests:** The authors declare no competing interests.

