## [Peer Review File · Nature Communications]

Reviewers' comments:

Reviewer #1 (Remarks to the Author):

This paper sought to understand an interesting phenomenon of meiotic recombination, where shorter chromosomes, which are at risk of not receiving enough DSBs to experience recombination, due to their size, in fact experience a higher DSB density than larger chromosomes. The authors discovered that this is partly due to the fact that a pathway, governed by the Pch2 ATPase and other factors, maintains a DSB formation potential in regions of fixed length located near the ends of chromosomes. This has the effect of having shorter chromosomes experiencing more DSBs.

The paper in general is well written and the data support well the conclusions, and will be of interest to the meiotic recombination community and in general people interested in chromosome biology.

However, there are a certain number of points that should be addressed, which are indicated below:

- Page 3, 2nd paragraph : « correlates well with DSB levels at a broad region scale »
 - Page 4, 2nd paragraph: "two distinct mechanisms", since we are not sure if they are fully independent at this stage.
 - Page 5, last line of 2nd paragraph: suggest "may be linked to chromosomal position".
 - Page 7 3rd paragraph: I found the fact that Mek1 is enriched that DSB hotspots quite striking (Figure S4). Could the authors comment about this, and mention if this has been observed before? Could this be a consequence of the *ndt80Δ* mutation or has this been detected in a *NDT80* background?
 - Page 8, 3rd paragraph: I think it is difficult to conclude from the ChIP seq data if the Hop1 signal is really decreased in the EARS, or if it looks below genome average because now interstitial regions are increased. It would be useful here to include a qPCR assay of Hop1 ChIP in *pch2 ndt80* versus *ndt80*, or clarify that the drop in the EARS is relative to the interstitial regions. Could the authors comment if these experiments imply that Hop1 is in limiting amounts?
 - Page 12, 1st paragraph: the chromosome size bias is still present when removing the EARS from the analysis, so the authors should specify that EARS are at least partially responsible for the late prophase enrichment of Hop1 on short chromosomes, and that other mechanisms are likely also participating to this bias.
 - Page 12, 2nd paragraph: it is not clear what the data in Figure 7F (*nup2Δ*) and 7G (*sir2Δ*) are compared to. If they want to emphasize the difference with NUP2 and SIR2, the authors should directly compare *nup2Δ* and NUP2 and *sir2Δ* and SIR2, at the 2 time-points for their statistical analysis (using the 3h and 6h *ndt80Δ* data).
 - Page 12, last line: "Zip1 polymerization" rather than "Zip1 deposition".
 - Page 27, last paragraph: indicate gel migration parameters in V/cm.
 - Page 9, end of second paragraph: are there also more frequent DSBs close to centromeres in the *pch2Δ Spo11* oligonucleotides experiment?
 - Page 9, 3rd paragraph, sentence "This effect, however, was due to diminished": here the difference between 3 and 6 h in interstitial regions in the *nup2Δ ndt80Δ* mutant should be compared to that in single *ndt80Δ* mutants.
- General comment on the data presentation: the layout of the graphs showing bootstrap-derived distributions is not sufficiently explained for a non-specialist. I understand that these graphs are generated using random samplings of a subset of the data of the same size as the studied regions (represented by a dot). Are these distributions supposed to represent the whole genome? And if yes, why is it in many instances so different from the genome average set to 1? It would help to remind briefly in the figures legends what these violin plots represent each time.

-It would be useful to provide in supplementary data a list of all the ChIP seq or Spo11 oligonucleotides datasets obtained for this study. I had a hard time to have an idea of all the datasets generated, especially for the Spo11 oligonucleotides.

Reviewer #2 (Remarks to the Author):

Small chromosomes are assured to receive at least one crossover per homolog pair, as required for homolog segregation. What makes smaller chromosomes special is a question that has intrigued geneticists for a long time. Using a combination of clever analyses, Subramanian and colleagues now provide a surprisingly simple and convincing answer: There is really nothing unique about small chromosomes. Rather, all end-proximal chromosome regions exhibit an increased ability to attract recombination events for an extended time interval. The fact that small chromosomes consist to a larger proportion of end-adjacent regions gives them an edge in attracting recombination initiation events. Moreover, the authors provide novel insights into the workings of three factors (Pch2, Sir2 Nup2) that control establishment or maintenance of chromosome end-associated recombination domains, putting a more mechanistic understanding of enhanced DSB formation within reach.

This is a great finding that will be of general biological interest. It is very appropriate for Nature Communications. The analysis is thorough, the wording is thoughtful, and the paper reads well. I really have only relatively minor points related to the data presentation.

Fig. 1A: Indicate in the Figure that ndt80D was analyzed

Fig. 1E: The y-axis labels are shifted.

Also: I repeatedly missed the yellow EAR dot when it was shifted substantially up which is the main discovery of this paper. You should use the arrangement from Fig. S3C where the boxed legend separates the chromosome diagram from the violin plot, or box the plot itself. Otherwise the yellow dot can easily be mistaken as part of the diagram. (Fig. 4G pch2ndt80 at t=6h is another confusing graph).

Fig. 4: It is unclear why sometimes "Repair intermediates" (4D), "DSBs" (4I) or "DSB + Repair intermediates" (S5) are quantitated, implying that DSBs are not included in 4D? I can understand that the cumulative recombination potential in ndt80 entails both DSBs and longer recombination intermediates. Why different combinations of these intermediates are quantitated in different figures needs to be explained. The authors should show the Southern blots for Fig. 4D in the supplement.

Fig. 4C,G: The reader has to do quite a bit of work to figure out why the violin plots for EAR- and CEN-proximal Hop1 enrichment look different. Could you indicate in the y-axis labeling that different bin sizes were used (2bk, 90kb??)?

Fig. 4F legend: "the grey dotted line" is repeated

Fig. 4H legend: Southern analysis of DSBs

Fig. 5: The figure title mentions DSBs, but no DSB data are shown in the Figure. The authors should add S6D, at least for nup2, to the main figure and show the Southern blots in the supplement.

Fig. 6A: Presence of overlap between Hop1 and Zip1 is difficult to see even in the zoomed version. Could you add a row for overlap and indicate plus and minus signs? I find one nucleus quite anecdotal without any indication of how many nuclei were examined, even if three nuclei are shown for nup2 sir2 ndt80. There are two options. Analyze a larger number of pachytene nuclei cytologically and indicate that number. Or show the Hop1 distribution in nup2 sir2 ndt80 by ChIP seq, as evidence that absence of Sir2 restores Hop1 distribution in nup1, and as ultimate validation of the pathway derived from limited cytological analysis. Is the Zip1 blob a polycomplex, and are those frequent in nup2?

Fig. S6F: The different Zip1 staining classes should all be represented in the same color in different genotypes to facilitate comparison.

Fig. 6B,C: The nup2ndt80 nuclei shown in 6B appear to have weaker nucleolar Pch2 staining than ndt80, in contrast to nuclei shown and quantitated in 6C. I understand that there is overlap between the weaker nup2 ndt80 and the stronger ndt80 nuclei, but it is acceptable practice to show representative nuclei rather than outliers.

Fig. 6D: The analysis of the pathway is very elegant, but I find the diagram overly simplistic. You should indicate that interstitial Hop1 spreading is controlled, since there potentially are other aspects of Hop1 (and Pch2/Sir2) function in meiotic progression that may be controlled differently and that are not analyzed here.

Fig. S1F, legend should be singular: only one bootstrap analysis is shown in a single violin plot

Reviewer #3 (Remarks to the Author):

Summary of relevance and findings:

Meiotic recombination creates new chromosomes in the germline and also ensure that homologous chromosomes segregate accurately at meiosis I. We have appreciated for a long time that meiotic recombination is increased on shorter chromosomes and that in many organisms, including human, crossovers are enriched near chromosome ends. However, we have no understanding of how this is regulated nor its functional significance.

In this work, Subramanian and colleagues present compelling evidence to suggest that double strand breaks (DSBs) that initiate meiotic recombination continue to accumulate during late prophase when interstitial regions become quiescent. They show that this is mediated by an enrichment of Hop1 near chromosome ends during late prophase. The spatio-temporal regulation of Hop1 association with chromosome end-adjacent regions (EARs) in late prophase depends on the AAA+-ATPase, Pch2, and intriguingly, accumulation of Zip1 at EARs does not appear to mediate removal of Hop1 in these regions (as it does in interstitial regions). Pch2 localisation is regulated by the nucleoporin Nup2 in a regulatory pathway the also includes inhibition of Sir2, histone deacetylase activity (Fig. 6D). Interestingly, the authors also identify EAR-like domains flanking the rDNA. The authors propose that Hop1 accumulation at EARs in late prophase ensures that all chromosomes receive at least one crossover and can explain why small chromosomes, where the contribution of EARs is proportionately greater, have higher recombination rates.

General comments:

The findings open up a new area of research that has eluded us for a long time and that may be relevant and conserved in other organisms. Although there are many questions that reviewers could ask the authors for, the manuscript is well-written and mature, consisting of a substantive and significant body of work that is appropriate for publication in the journal. The figures and data are of a high quality, experiments are well-controlled such as the inclusion of spo11-deletion controls in ChIP-seq experiments as well as reproducibility of the major findings in wild type cells (e.g. Fig. 1F), ruling out artefacts due to ndt80 arrest.

I do have one comment about the general discussion of the significance of EARs, which is that the authors should include a comment about some plant species (e.g. wheat), where early replication near chromosome ends correlated with elevated levels of DSBs early. This is a different mechanism that would also ensure an obligate crossover. It would therefore also be pertinent for the model that EARs ensure obligate crossovers to include in the discussion that this spatiotemporal replication is not conserved in budding yeast.

On page 8, the authors use the term long-lived hotspots. I am not sure this term is precise. What is meant is that hotspots are used in late prophase as well as earlier stages, giving rise to a DSB

(and recombination intermediate signal) throughout the time course.

In Figures 4H and I, how do the authors distinguish between persistent DSB activity (text, p. 9) as opposed to repair that is blocked?

Normally, Red1 is thought to recruit Hop1 to chromatin. However, unlike Hop1, Red1 levels do not increase in EARs in late prophase. The conclusion that Hop1 accumulation occurs by different means since Red1 chromosome bias did not increase (Figure S7C, p. 12) relies on the assumption that Red1 is not in excess (along the entire genome). Perhaps the authors can comment and review their statement, pending whether they think this is possible?

The Discussion is thorough but not repetitive. Perhaps it would be worthwhile re-emphasizing again that Ndj1 and Nup2 did have phenotypes (lack of Hop1 accumulation in EARs), but this was likely due to a failure arising from synapsis defects or failure to remove Hop1 from interstitial sites?

Minor comments:

Please refer to a figure or reference on p. 13 in Discussion that you observe centromeric DSBs in early prophase.

Some of the colors are a bit hard to follow in the figures. Especially when shades of the same color is used and the shapes (i.e. circles) are the same. Can you please use different shapes or something like that?

We would like to thank the reviewers for their enthusiasm and constructive comments, which helped to further improve this manuscript.

Reviewer #1 (Remarks to the Author):

This paper sought to understand an interesting phenomenon of meiotic recombination, where shorter chromosomes, which are at risk of not receiving enough DSBs to experience recombination, due to their size, in fact experience a higher DSB density than larger chromosomes. The authors discovered that this is partly due to the fact that a pathway, governed by the Pch2 ATPase and other factors, maintains a DSB formation potential in regions of fixed length located near the ends of chromosomes. This has the effect of having shorter chromosomes experiencing more DSBs.

The paper in general is well written and the data support well the conclusions, and will be of interest to the meiotic recombination community and in general people interested in chromosome biology.

However, there are a certain number of points that should be addressed, which are indicated below:

*-Page 3, 2nd paragraph : « correlates well with DSB levels at a broad region scale »
- Has been corrected.*

*-Page 4, 2nd paragraph: “two distinct mechanisms”, since we are not sure if they are fully independent at this stage.
- Has been corrected.*

*-Page 5, last line of 2nd paragraph: suggest “may be linked to chromosomal position”.
- Has been corrected.*

*-Page 7 3rd paragraph: I found the fact that Mek1 is enriched that DSB hotspots quite striking (Figure S4). Could the authors comment about this, and mention if this has been observed before? Could this be a consequence of the *ndt80*Δ mutation or has this been detected in a *NDT80* background?*

*- Mek1-activity dependent phosphorylated H3T11 is also observed at DSB sites (Kniewel et al. Genetics 2017). We have added this note in the text. ChIP-seq for H3T11ph was done in *NDT80* cells so the pattern of Mek1 at DSB sites is unrelated to *ndt80* deletion.*

*-Page 8, 3rd paragraph: I think it is difficult to conclude from the ChIP seq data if the Hop1 signal is really decreased in the EARS, or if it looks below genome average because now interstitial regions are increased. It would be useful here to include a qPCR assay of Hop1 ChIP in *pch2 ndt80* versus *ndt80*, or clarify that the drop in the EARs is relative to the interstitial regions. Could the authors comment if these experiments imply that Hop1 is in limiting amounts?*

- We have now included quantitative ChIP-seq analysis using spike-in controls for Hop1 in *ndt80* (Figs. 2f and S2e) and *pch2 ndt80* (Fig. 4d). These data show that Hop1 levels on chromosomes decrease overall between early and late/extended prophase, but increase in *pch2* mutants over time, consistent with cytological analyses. Our results indicate that Hop1 persists in the EARs in late prophase and that further Hop1 accumulates in the interstitial regions in *pch2* mutants.

Our data do not allow us to conclude whether or not Hop1 is limiting. Although the elevated Hop1 levels in *pch2* mutants could argue that Hop1 is not limiting, the Burgess lab (Ho and Burgess, 2011) previously showed that *pch2* mutants have elevated levels of total Hop1 protein. Thus, it is still possible that Hop1 is in limiting amounts. We included a brief note about the elevated Hop1 protein levels in *pch2* mutants in the text.

-Page 12, 1st paragraph: *the chromosome size bias is still present when removing the EARs from the analysis, so the authors should specify that EARs are at least partially responsible for the late prophase enrichment of Hop1 on short chromosomes, and that other mechanisms are likely also participating to this bias.*

- Has been corrected.

-Page 12, 2nd paragraph: *it is not clear what the data in Figure 7F (*nup2Δ*) and 7G (*sir2Δ*) are compared to. If they want to emphasize the difference with NUP2 and SIR2, the authors should directly compare *nup2Δ* and NUP2 and *sir2Δ* and SIR2, at the 2 time-points for their statistical analysis (using the 3h and 6h *ndt80Δ* data).*

- In Fig. 7f and 7g, we now additionally compare *nup2 ndt80* and *sir2 ndt80* to *ndt80* for the 3 hr and 6hr time points.

-Page 12, last line: *“Zip1 polymerization” rather than “Zip1 deposition”.*

- We acknowledge that our data probably indicate Zip1 polymerization but ChIP-seq (a population assay) cannot evaluate polymerization along each chromosome. Hence, we prefer to say “deposition” to more accurately describe our findings.

-Page 27, last paragraph: *indicate gel migration parameters in V/cm.*

- Has been corrected.

-Page 9, end of second paragraph: *are there also more frequent DSBs close to centromeres in the *pch2Δ Spo11* oligonucleotides experiment?*

- Yes. We are now included a plot (Supplementary Fig. 5e) showing Spo11-oligo counts around centromeres.

-Page 9, 3rd paragraph, sentence *“This effect, however, was due to diminished”*: *here the difference between 3 and 6 h in interstitial regions in the *nup2Δ ndt80Δ* mutant should be compared to that in single *ndt80Δ* mutants.*

- Has been corrected.

-General comment on the data presentation: *the layout of the graphs showing bootstrap-derived distributions is not sufficiently explained for a non-specialist. I understand that these*

graphs are generated using random samplings of a subset of the data of the same size as the studied regions (represented by a dot). Are these distributions supposed to represent the whole genome? And if yes, why is it in many instances so different from the genome average set to 1? It would help to remind briefly in the figures legends what these violin plots represent each time.

- Reviewer #2 raised a similar point. The reason why the distributions are often different from genome average is related to the structure of the data and the size of the sampling bins. In particular large sampling bins (e.g. 2x90 kb for the EARs) will often at least partially overlap with the EARs because EARs overall occupy about 24% of the non-repetitive yeast genome. Indeed, because we sample two EAR-size regions on each of the 16 chromosomes, the samples on the short chromosomes will almost always also capture the EARs. With smaller bins (2 kb for the CENs), this effect is much more limited because the chance of not overlapping with an EAR becomes substantially higher. Following the suggestion of reviewer #2, we have now labeled the y-axis with bin-size information and also added this information in the figure legends.

-It would be useful to provide in supplementary data a list of all the ChIP seq or Spo11 oligonucleotides datasets obtained for this study. I had a hard time to have an idea of all the datasets generated, especially for the Spo11 oligonucleotides.

- We have now added Supplementary tables 4 and 5 listing the ChIP-seq and Spo11-oligo datasets respectively.

Reviewer #2 (Remarks to the Author):

Small chromosomes are assured to receive at least one crossover per homolog pair, as required for homolog segregation. What makes smaller chromosomes special is a question that has intrigued geneticists for a long time. Using a combination of clever analyses, Subramanian and colleagues now provide a surprisingly simple and convincing answer: There is really nothing unique about small chromosomes. Rather, all end-proximal chromosome regions exhibit an increased ability to attract recombination events for an extended time interval. The fact that small chromosomes consist to a larger proportion of end-adjacent regions gives them an edge in attracting recombination initiation events. Moreover, the authors provide novel insights into the workings of three factors (*Pch2*, *Sir2*, *Nup2*) that control establishment or maintenance of chromosome end-associated recombination domains, putting a more mechanistic understanding of enhanced DSB formation within reach.

This is a great finding that will be of general biological interest. It is very appropriate for Nature Communications. The analysis is thorough, the wording is thoughtful, and the paper reads well. I really have only relatively minor points related to the data presentation.

Fig. 1A: Indicate in the Figure that *ndt80D* was analyzed
- Has been corrected.

Fig. 1E: The y-axis labels are shifted.
- Has been corrected.

Also: I repeatedly missed the yellow EAR dot when it was shifted substantially up which is the main discovery of this paper. You should use the arrangement from Fig. S3C where the boxed legend separates the chromosome diagram from the violin plot, or box the plot itself. Otherwise the yellow dot can easily be mistaken as part of the diagram. (Fig. 4G *pch2ndt80* at *t=6h* is another confusing graph).

- Thank you for this suggestion. This has been corrected.

Fig. 4: It is unclear why sometimes "Repair intermediates" (4D), "DSBs" (4I) or "DSB + Repair intermediates" (S5) are quantitated, implying that DSBs are not included in 4D? I can understand that the cumulative recombination potential in *ndt80* entails both DSBs and longer recombination intermediates. Why different combinations of these intermediates are quantitated in different figures needs to be explained. The authors should show the Southern images for Fig. 4D in the supplement.

- We have now included the following note about this in the methods section. "In *ndt80Δ* samples, repair intermediates provide a cumulative view of total DSB occurrence. Thus, in Southern images where DSB signals were faint (**Supplementary Fig. 3g**), the sum of DSBs and repair intermediates was quantified. Only repair intermediates were quantified in **Fig. 4e, 6d**, because DSB signals were very faint and could not be quantified reliably."
We have now included the Southern images in Supplementary Fig. 5.

Fig. 4C,G: The reader has to do quite a bit of work to figure out why the violin plots for EAR-

and CEN-proximal Hop1 enrichment look different. Could you indicate in the y-axis labeling that different bin sizes were used (2bk, 90kb??)?

- Thank you for this suggestion. We have now labeled the y-axis with bin-size information and also added this information in the figure legends. We also included a brief explanation in the legend of Figure 1 for why the violin plots in the larger sample bins do not center on the genome average (see also response to a similar point by reviewer #1).

Fig. 4F legend: "the grey dotted line" is repeated

- Has been corrected.

Fig. 4H legend: Southern analysis of DSBs

- Has been corrected.

Fig. 5: The figure title mentions DSBs, but no DSB data are shown in the Figure. The authors should add S6D, at least for *nup2*, to the main figure and show the Southern blots in the supplement.

- We now show this data in the main figures (Fig. 6d) and we have added Southern blot images to Supplementary Fig. 5 a, b.

Fig. 6A: Presence of overlap between Hop1 and Zip1 is difficult to see even in the zoomed version. Could you add a row for overlap and indicate plus and minus signs? I find one nucleus quite anecdotal without any indication of how many nuclei were examined, even if three nuclei are shown for *nup2 sir2 ndt80*. There are two options. Analyze a larger number of pachytene nuclei cytologically and indicate that number. Or show the Hop1 distribution in *nup2 sir2 ndt80* by ChIP seq, as evidence that absence of Sir2 restores Hop1 distribution in *nup1*, and as ultimate validation of the pathway derived from limited cytological analysis. Is the Zip1 blob a polycomplex, and are those frequent in *nup2*?

- We have now added a row to indicate overlap between Hop1 and Zip1. For each genotype, $n > 40$ nuclei were analyzed. We have included this information in the figure legend. Yes, the blob of Zip1 is likely a polycomplex, which is common in *ndt80* mutants and slightly more common in *nup2 ndt80* mutants. The Burgess lab previously also showed polycomplex formation in *nup2* mutants using time-lapse analysis of SC formation (Chu et al 2017). However, they found only a very minor delay in SC assembly, consistent with our observations.

Fig. S6F: The different Zip1 staining classes should all be represented in the same color in different genotypes to facilitate comparison.

- Has been corrected.

Fig. 6B,C: The *nup2ndt80* nuclei shown in 6B appear to have weaker nucleolar Pch2 staining than *ndt80*, in contrast to nuclei shown and quantitated in 6C. I understand that there is overlap between the weaker *nup2 ndt80* and the stronger *ndt80* nuclei, but it is acceptable practice to show representative nuclei rather than outliers.

- We have now replaced the images with representative nuclei.

Fig. 6D: The analysis of the pathway is very elegant, but I find the diagram overly simplistic.

You should indicate that interstitial Hop1 spreading is controlled, since there potentially are other aspects of Hop1 (and Pch2/Sir2) function in meiotic progression that may be controlled differently and that are not analyzed here.

- We have now labeled Hop1 as “interstitial Hop1”

Fig. S1F, legend should be singular: only one bootstrap analysis is shown in a single violin plot

- Has been corrected.

Reviewer #3 (Remarks to the Author):

Summary of relevance and findings:

Meiotic recombination creates new chromosomes in the germline and also ensure that homologous chromosomes segregate accurately at meiosis I. We have appreciated for a long time that meiotic recombination is increased on shorter chromosomes and that in many organisms, including human, crossovers are enriched near chromosome ends. However, we have no understanding of how this is regulated nor its functional significance.

In this work, Subramanian and colleagues present compelling evidence to suggest that double strand breaks (DSBs) that initiate meiotic recombination continue to accumulate during late prophase when interstitial regions become quiescent. They show that this is mediated by an enrichment of Hop1 near chromosome ends during late prophase. The spatio-temporal regulation of Hop1 association with chromosome end-adjacent regions (EARs) in late prophase depends on the AAA+-ATPase, Pch2, and intriguingly, accumulation of Zip1 at EARs does not appear to mediate removal of Hop1 in these regions (as it does in interstitial regions). Pch2 localisation is regulated by the nucleoporin Nup2 in a regulatory pathway the also includes inhibition of Sir2, histone deacetylase activity (Fig. 6D). Interestingly, the authors also identify EAR-like domains flanking the rDNA. The authors propose that Hop1 accumulation at EARs in late prophase ensures that all chromosomes receive at least one crossover and can explain why small chromosomes, where the contribution of EARs is proportionately greater, have higher recombination rates.

General comments:

The findings open up a new area of research that has eluded us for a long time and that may be relevant and conserved in other organisms. Although there are many questions that reviewers could ask the authors for, the manuscript is well-written and mature, consisting of a substantive and significant body of work that is appropriate for publication in the journal. The figures and data are of a high quality, experiments are well-controlled such as the inclusion of spo11-deletion controls in ChIP-seq experiments as well as reproducibility of the major findings in wild type cells (e.g. Fig. 1F), ruling out artefacts due to ndt80 arrest.

I do have one comment about the general discussion of the significance of EARs, which is that the authors should include a comment about some plant species (e.g. wheat), where early replication near chromosome ends correlated with elevated levels of DSBs early. This is a different mechanism that would also ensure an obligate crossover. It would therefore also be pertinent for the model that EARs ensure obligate crossovers to include in the discussion that this spatiotemporal replication is not conserved in budding yeast.

- Thank you for this suggestion. We now refer to work by the Franklin lab in barley on early recombination initiation in the subtelomeric regions.

On page 8, the authors use the term long-lived hotspots. I am not sure this term is precise.

What is meant is that hotspots are used in late prophase as well as earlier stages, giving rise to a DSB (and recombination intermediate signal) throughout the time course.

- We have now added to our introduction (Page 4, last line of first paragraph) a more precise explanation of the terminology “long-lived hotspots” that is used throughout the manuscript.

In Figures 4H and I, how do the authors distinguish between persistent DSB activity (text, p. 9) as opposed to repair that is blocked?

- We know the DSB activity is elevated because the elevated DSBs are accompanied by an increase in repair intermediates. We now mention this in the text.

Normally, Red1 is thought to recruit Hop1 to chromatin. However, unlike Hop1, Red1 levels do not increase in EARs in late prophase. The conclusion that Hop1 accumulation occurs by different means since Red1 chromosome bias did not increase (Figure S7C, p. 12) relies on the assumption that Red1 is not in excess (along the entire genome). Perhaps the authors can comment and review their statement, pending whether they think this is possible?

- We were not expecting a further increase of Red1 in the EARs because Red1 remains associated with synapsed chromosomes and does not experience the strong SC-dependent removal from the interstitial regions seen for Hop1. We now mention this point in the text.

The Discussion is thorough but not repetitive. Perhaps it would be worthwhile re-emphasizing again that Ndj1 and Nup2 did have phenotypes (lack of Hop1 accumulation in EARs), but this was likely due to a failure arising from synapsis defects or failure to remove Hop1 from interstitial sites?

- We have included this point in the discussion.

Minor comments:

Please refer to a figure or reference on p. 13 in Discussion that you observe centromeric DSBs in early prophase.

- We have now added the figure reference.

Some of the colors are a bit hard to follow in the figures. Especially when shades of the same color is used and the shapes (i.e. circles) are the same. Can you please use different shapes or something like that?

- We are now using squares to represent the 3hr samples.

REVIEWERS' COMMENTS:

Reviewer #1 (Remarks to the Author):

The authors have satisfactorily answered to all my points.
To me the manuscript is now suitable for publication.